# Endothelial gene regulatory elements associated with cardiopharyngeal lineage differentiation
Ilaria Aurigemma[1,2], Olga Lanzetta[3], Andrea Cirino[3], Sara Allegretti[1], Gabriella Lania[3], Rosa Ferrentino[3], Varsha Poondi Krishnan[3], Claudia Angelini[4], Elizabeth Illingworth[2] & Antonio Baldini ⓘ[1,5] ✉

Endothelial cells (EC) differentiate from multiple sources, including the cardiopharyngeal mesoderm, which gives rise also to cardiac and branchiomeric muscles. The enhancers activated during endothelial differentiation within the cardiopharyngeal mesoderm are not completely known. Here, we use a cardiogenic mesoderm differentiation model that activates an endothelial transcription program to identify endothelial regulatory elements activated in early cardiogenic mesoderm. Integrating chromatin remodeling and gene expression data with available single-cell RNA-seq data from mouse embryos, we identify 101 putative regulatory elements of EC genes. We then apply a machine-learning strategy, trained on validated enhancers, to predict enhancers. Using this computational assay, we determine that 50% of these sequences are likely enhancers, some of which are already reported. We also identify a smaller set of regulatory elements of well-known EC genes and validate them using genetic and epigenetic perturbation. Finally, we integrate multiple data sources and computational tools to search for transcriptional factor binding motifs. In conclusion, we show EC regulatory sequences with a high likelihood to be enhancers, and we validate a subset of them using computational and cell culture models. Motif analyses show that the core EC transcription factors GATA/ETS/FOS is a likely driver of EC regulation in cardiopharyngeal mesoderm.

## Background

In mammals, endothelial cells (EC) derive, through vasculogenesis, from the mesoderm but they can differentiate from multiple sources[1,2]. ECs are heterogeneous in their function, transcriptional program, and chromatin landscape; single-cell-based studies have provided a wealth of data on this effect[3,4]. Heterogeneity has different causes[5], but, at least in part, it may depend upon lineage of origin as well as epigenomic and enhancer profiles.

The cardiopharyngeal mesoderm (CPM) lineage provides progenitors to various tissues and organs of the lower face, mediastinum, and heart[6]. The CPM also provides multipotent progenitors that differentiate into ECs[7–10]. In addition, the second heart field (SHF)[11–13] which derives from the CPM, provides EC progenitors to various components of the cardiovascular system, including ECs of the pharyngeal arch arteries and outflow tract[14,15], which, through endothelial-to-mesenchymal transition, contribute to cardiac valve formation. In particular, *Tbx1* expression, which is a marker of the CPM and SHF, identified these ECs through genetic labeling driven by a *Tbx1*[Cre] allele[15,16]. In addition, time-controlled genetic labeling with an inducible Cre recombinase determined that *Tbx1* was activated in EC progenitors within the time window E7.5-E8.5 in mouse embryos[15]. The molecular events that drive EC differentiation in the CPM are mostly unknown, with some exception[8], and a suitable approach to define them would be to use a dynamic model in which chromatin remodeling and gene expression can be monitored at critical developmental times. Using such an approach, we found that the differentiation of cardiogenic mesoderm from mouse embryonic stem cells (mESCs) activated an endothelial transcription program. We then measured gene expression and chromatin accessibility in differentiating mESCs within the activation window to identify differentially expressed genes and differentially accessible regions. We then used EC gene

[1]PhD program in Molecular Medicine and Medical Biotechnology, University Federico II, Via Sergio Pansini 5, 80131 Naples, Italy. [2]Department of Chemistry and Biology, University of Salerno, Via Giovanni Paolo II 132, 84084 Fisciano, Italy. [3]Institute of Genetics and Biophysics, National Research Council, Via Pietro Castellino 111, 80131 Naples, Italy. [4]Istituto Applicazioni del Calcolo, National Research Council, Via Pietro Castellino 111, 80131 Naples, Italy. [5]Department of Molecular Medicine and Medical Biotechnology, University Federico II, Via Sergio Pansini 5, 80131 Naples, Italy. ✉e-mail: antonio.baldini@unina.it

information from published single-cell RNA-seq data obtained from *Tbx1^Cre* and *Mesp1^Cre* sorted cells from E8.5-E9.5 mouse embryos. Data integration and analysis identified and computationally scored 101 putative regulatory elements activated in the *Tbx1^Cre*-selected EC cluster. Finally, we identified and validated putative regulatory elements associated with a small set of well-known EC genes.

In summary, our results provide a systematic experimental approach to identify cell type-specific regulatory elements during differentiation, and the results obtained shed light on the EC regulatory elements activated during cardiogenic mesoderm differentiation.

## Results

### Cardiac mesoderm differentiation of mouse embryonic stem cells activates endothelial differentiation, and it can be driven to yield a nearly homogeneous EC population

We have used a published protocol to derive cardiac mesoderm from mouse embryonic stem cells (mESCs)[17] (Fig. 1a). We found expression of endothelial genes at differentiation day 4 (d4), while these genes were not detected at d2 (Fig. 1b). RNA-seq analysis performed on two replicates from two independent differentiation experiments confirmed the activation of an endothelial expression program within this time window (Fig. 1c, Table 1, and Supplementary Data 1). Therefore, we used the d2-d4 time window in our search for EC enhancers.

Flow cytometry using the endothelial-specific marker VE-Cadherin, encoded by the *Cdh5* gene, revealed that at d2, there were no detectable VE-Cadherin+ cells, while at d4, a small percentage (19%) was present (Fig. 2a). Therefore, we extended the differentiation protocol in order to increase the EC population. To this end, at d4, we added a high concentration of VEGFA

(200 ng/ml) and Forskolin (2 μM), as suggested by a published protocol[18]. Following this treatment, at d6 and d8, the percentage of VE-Cadherin+ cells increased to 57.4 and 91.3%, respectively (Fig. 2a). Similar results were obtained in multiple experiments. Next, we performed Matrigel assays on d8 cells[19] in order to determine whether they formed the tubule-like networks expected of fully differentiated ECs. Results indicated that d8 cells had this capacity (Fig. 2b). Thus, this modified differentiation protocol produced a nearly homogeneous population of ECs.

### An unbiased strategy identifies putative regulatory elements in early EC differentiation

Having identified the d2-d4 time window for the activation of an EC transcription program, we performed ATAC-seq at these two time points in order to localize regions of dynamic chromatin accessibility across the genome. Experiments were performed in two biological replicates (from two independent differentiation experiments), and we subsequently considered consensus peaks only, i.e., peaks that were called in both replicates. Thus, we identified a total of 20,268 consensus peaks at d2 and 17,110 at d4 (Supplementary Data 2), of which 8773 were differentially accessible regions (DARs) as determined by the DiffBind and Descan2 software tools[20,21]; again, we only considered DARs that were identified by both tools (Fig. 3a, b). We then derived a list of marker genes of an EC cluster obtained from single-cell RNA-seq (scRNA-seq) experiments performed on cells selected using a *Tbx1Cre* driver combined with a GFP reporter. *Tbx1* is a CPM marker, and cells were FACS-purified from E8.5 and E9.5 mouse embryos[22]. The EC cluster was characterized by 252 marker genes (listed in Supplementary Data 3) that we intersected with our dataset of d2-d4 DARs opened at d4 (n = 4408) (Fig. 3a). This resulted in 101 regions that were significantly

**Fig. 1 | Activation of an EC transcription program during cardiogenic mesoderm differentiation of mESC. a** Schematic illustration of the EC differentiation protocol from mESCs. **b** Expression of pluripotency (*Oct3/4; Nanog; Rex1*), mesodermal (*Brachyury; Mesp1; Pdgfrα; Gata4*), endothelial marker genes (*Pecam1; Eng; Kdr; Cdh5; Nos3; Flt1; Gata6; Notch1*), and the CPM marker *Tbx1* during differentiation by RT-PCR. *Gapdh* was used as a normalizer. The molecular weight marker is the 100 bp ladder. Uncropped photographs of gels are reported in Supplementary Fig. 1. **c** RNA-seq volcano plot of differentially expressed genes (DEGs) in d4 vs d2 samples in two biological replicates. Genes downregulated at d4 (n. 731) are indicated in blue and genes upregulated are in red (n. 1088). We indicate examples of endothelial marker genes (*Flt1; Kdr; Ets1; Fli1; Gata4; Pecam1; Cdh5; Etv2; Gata6; Gata1; Erg; Tie1; Gata2, Notch1*).

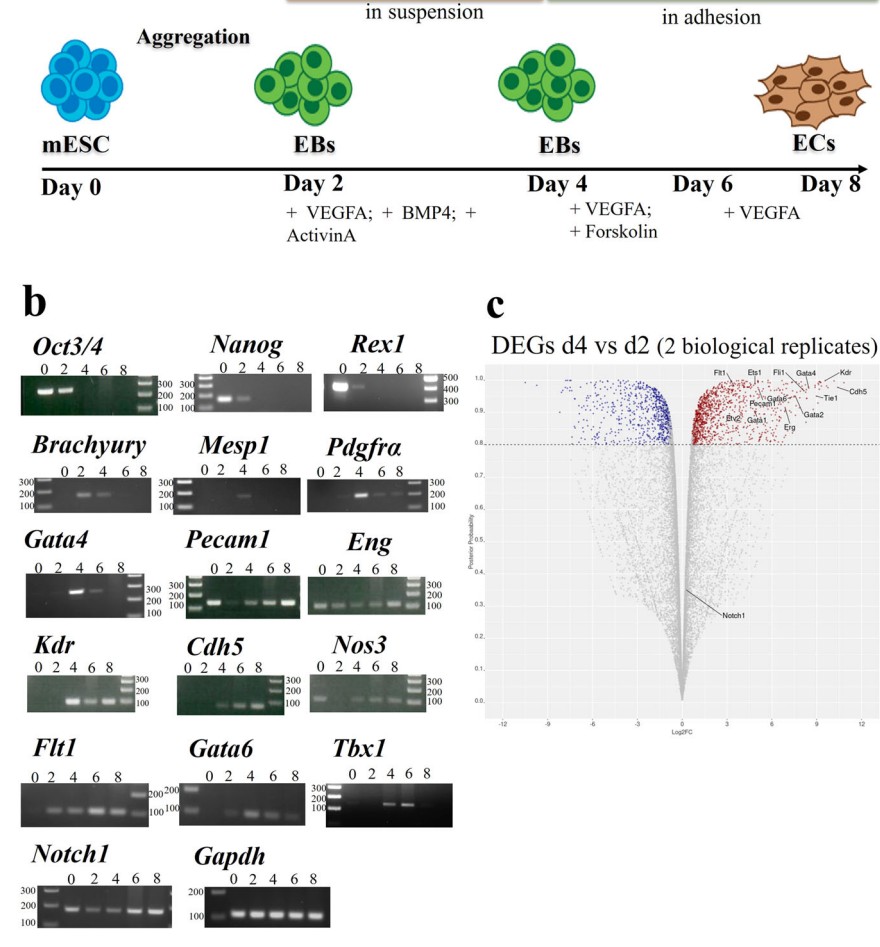

more accessible at d4, compared to d2, and were associated with EC expressed genes (Table 2). We then performed a computational prediction of the probability that these regions are enhancers. To this end, we used a machine-learning procedure based on a logistic regression test trained on validated enhancer sequences (see methods for details). The average scores obtained with this procedure are reported in Table 2. Of the 101 regions

**Table 1 | Gene ontology of genes significantly upregulated at d4 compared to d2**

| Source | Term | Term id | adj *p* value | Term size | Query size | Intersection |
|---|---|---|---|---|---|---|
| GO:BP | anatomical structure morphogenesis | GO:0009653 | 1.84E-52 | 1999 | 1082 | 353 |
| GO:BP | circulatory system development | GO:0072359 | 8.39E-52 | 900 | 1082 | 217 |
| GO:BP | anatomical structure development | GO:0048856 | 5.11E-50 | 4047 | 1082 | 549 |
| GO:BP | multicellular organismal process | GO:0032501 | 1.56E-49 | 4609 | 1082 | 597 |
| GO:BP | anatomical structure formation involved in morphogenesis | GO:0048646 | 1.75E-48 | 877 | 1082 | 208 |
| GO:BP | **vasculature development***  | GO:0001944 | 7.91E-48 | 583 | 1082 | 164 |
| GO:BP | cell surface receptor signaling pathway | GO:0007166 | 1.37E-47 | 1786 | 1082 | 318 |
| GO:BP | **blood vessel development***  | GO:0001568 | 2.07E-47 | 556 | 1082 | 159 |
| GO:BP | developmental process | GO:0032502 | 2.07E-47 | 4400 | 1082 | 573 |
| GO:BP | signal transduction | GO:0007165 | 1.35E-45 | 3507 | 1082 | 488 |

* GO terms most relevant to this work.

**Fig. 2 | Progressive EC differentiation from cardiogenic mesoderm. a** Flow cytometry using anti-VE-Cadherin antibody during mESC differentiation. The VE-Cadherin⁺ subpopulation is identified at days 4-6-8 of differentiation. The negative control is isotype IgG1 control antibody-labeled differentiating cells. **b** In vitro tube formation assay (Matrigel) of d4 cells (left, negative control) and d8 cells (right) plated for 24 h. The scale bar is 100 μm.

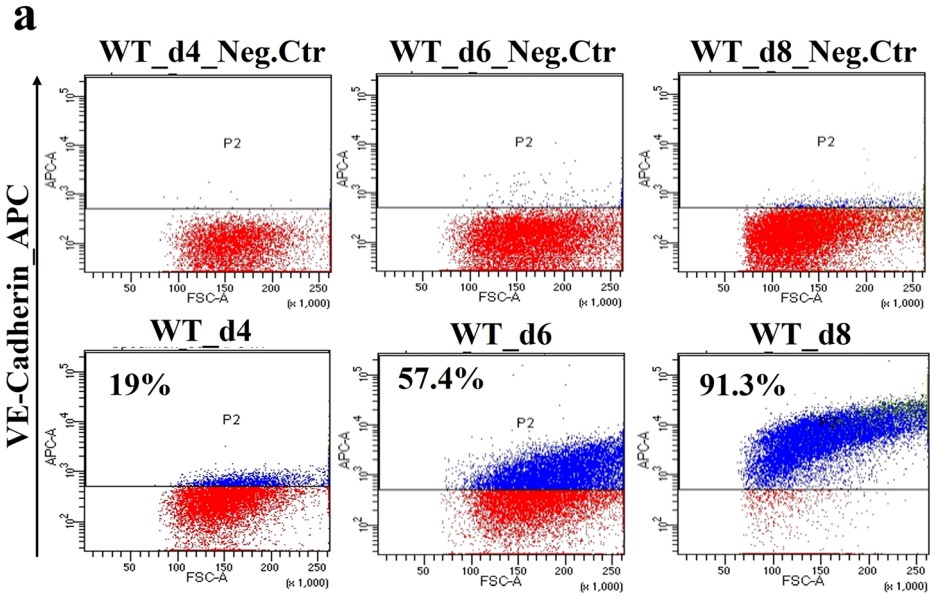

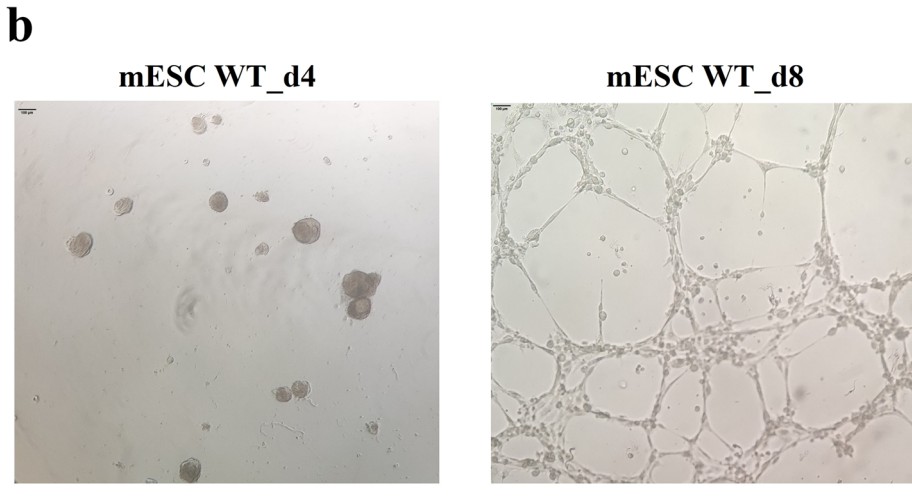

**Fig. 3 | Chromatin remodeling during mESC differentiation. a** Volcano plot of differentially accessible regions (DARs) in d4 vs d2 samples. In blue are DARs decreased at day 4; in red are DARs increased at day 4. **b** Distribution of total ATAC peaks at d2, d4, and DARs relative to gene features. The promoter region has been set at ±1000 bp to the transcription start site (TSS). Data sources are shown in Supplementary Data 6. **c** Enriched known motifs evaluated by HOMER using DARs mapped to marker genes of the EC cluster reported by Nomaru et al., 2021 selected from the *Tbx1^{Cre}*-sorted population of mouse embryos at E8.5 and E9.5. The full motif search results are reported in Supplementary Data 4.

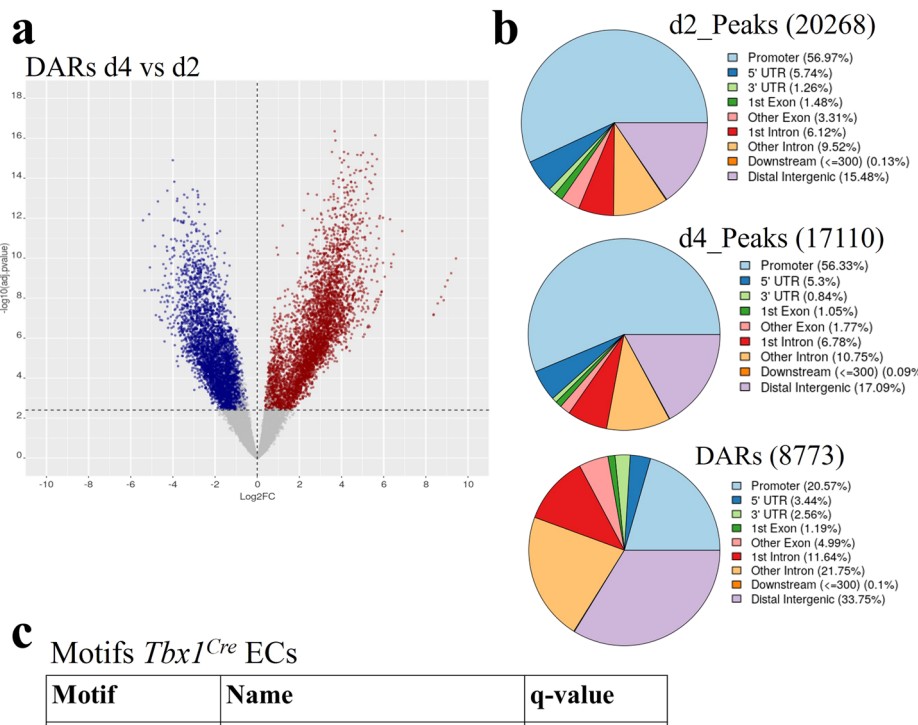

identified, 57 scored more than 0.5, indicating a significant likelihood of being enhancers; of these, 15 (26%) have been reported in the literature (references indicated in Table 2). A search for transcription factor binding motifs within the 101 regions identified significantly enriched motifs (as the background, we used the peakome associated with expressed genes). Specifically, we found motifs of GATA, ETS transcription factor families, and FOS, a subunit of the AP-1 transcription complex (Fig. 3c and Supplementary Data 4). GATA and ETS factors are co-present in 55% (56 out of 101) of the regions tested. The expression of *Gata1, Gata2, Gata4, Gata6, Fos, and Erg*, as well as other ETS family members (e.g., *Ets1, Ets2, Etv2, Fli1, Elk1, Elf1*) genes were strongly upregulated at d4, relative to d2 (Supplementary Data 1).

Next, we repeated the procedure using marker genes expressed in endothelial clusters of a scRNA-seq dataset from *Mesp1^{Cre}*-sorted cells at the same developmental stage (E9.5)[22]. *Mesp1^{Cre}*-sorted cells include ECs derived from the entire anterior mesoderm and not just the cardiopharyngeal mesoderm, practically the entire vascular bed of the trunk, as the head and most posterior regions of the embryos were removed before sorting[22]. In this study, two endothelial clusters were identified, named c2

and c16, that shared 801 marker genes (Supplementary Data 3). We mapped DARs upregulated at d4 to these sets of genes and identified 536 unique putative regulatory elements, of which, 283 (52.8%) had a score above 0.5 (Supplementary Data 3). We conducted motif searches using DARs upregulated at d4 mapped to marker genes of the two endothelial clusters separately, 434 regions for *Mesp1^{Cre}* c16 and 367 regions for *Mesp1^{Cre}* c2. Results identified a more extensive set of motifs than the one identified using the *Tbx1^{Cre}* dataset, but the most enriched ones were again GATA and ETS factors (Supplementary Data 4).

### Identification and validation of EC regulatory elements (RE) associated with major EC differentiation genes

Next, we applied a different approach to the identification of EC-REs: we selected a group of well-known endothelial genes that were expressed at d4, on the basis of RNA-seq data, and that exhibited regions of increased chromatin accessibility at d4. We focused on 6 putative, non-promoter REs associated with six: *Kdr* (encoding VEGFR2), *Chd5* (encoding VE-CADHERIN), *Eng* (encoding ENDOGLIN), *Flt1* (encoding VEGFR1), *Pecam1*, and *Notch1*. Computational prediction indicated that four of the six

**Article**

**Table 2 | Chromatin regions opened at d4 and mapped to EC marker genes of a *Tbx1^Cre* - selected EC cluster**

| Genome coordinates* | logFC (ATAC) | P value | FDR | Location | Distance To TSS | Gene | Log2FC (gene exp) | Prediction score | Reference |
|---|---|---|---|---|---|---|---|---|---|
| chr11-55022942-55023482 | 1.731305088 | 0.000271592 | 0.001006387 | intron 1 of 24 | 9963 | Anxa6 | 1.840085406 | 0.991523828 | |
| chr7-99238337-99239019 | 1.309011817 | 0.001473239 | 0.004457452 | Promoter | 0 | Mogat2 | 3.443899948 | 0.985975573 | |
| chr5-30913178-30913820 | 2.044580514 | 4.92016E-06 | 3.10012E-05 | Promoter | 0 | Emilin1 | 3.926143195 | 0.980203915 | |
| chr10-21936372-21937389 | 1.727703347 | 1.31823E-05 | 7.14788E-05 | intron 2 of 13 | 54188 | Sgk1 | -0.757368697 | 0.979240582 | 48 |
| chr13-89540219-89540775 | 1.579104843 | 0.000127698 | 0.000514763 | Promoter | 423 | Hapln1 | 8.244174051 | 0.968024995 | 49 |
| chr16-11063157-11063734 | 2.122247938 | 1.95731E-05 | 0.000100342 | intron 1 of 1 | 2423 | Litaf | 0.807920379 | 0.963443838 | |
| chr8-84978495-84979128 | 0.861368199 | 0.002006188 | 0.005864294 | Promoter | 0 | Junb | 3.012397503 | 0.960556135 | |
| chr2-148443115-148443933 | 2.029241109 | 8.16186E-06 | 4.73763E-05 | Promoter | 0 | Cd93 | 5.184191371 | 0.955478741 | |
| chr3-115641286-115641901 | 4.54895537 | 3.88153E-13 | 1.0221E-10 | Distal Intergenic | 73171 | S1pr1 | 6.078338735 | 0.949913648 | |
| chr16-38653136-38654086 | 3.265019366 | 3.16978E-10 | 1.38153E-08 | intron 1 of 11 | 59188 | Arhgap31 | 4.096226015 | 0.946065518 | 50 |
| chr2-84424961-84425679 | 1.371286656 | 8.62775E-06 | 4.97039E-05 | Promoter | 0 | Calcrl | 5.437845621 | 0.942004327 | |
| chr3-115642058-115643118 | 1.157937743 | 0.001271082 | 0.003907539 | Distal Intergenic | 71954 | S1pr1 | 6.078338735 | 0.937703732 | |
| chr6-136930481-136931295 | 2.262313246 | 1.46906E-08 | 2.73155E-07 | intron 2 of 4 | 10604 | Arhgdib | 4.609108878 | 0.924445706 | |
| chr2-181675354-181675918 | 1.101504454 | 0.001247142 | 0.003843758 | Distal Intergenic | -3714 | Sox18 | 6.278620495 | 0.920778081 | |
| chrX-155215770-155216892 | 0.697063226 | 0.000500851 | 0.00172564 | Promoter | 0 | Sat1 | 0.38385953 | 0.913155221 | |
| chr17-35051612-35052273 | 1.301729513 | 0.000104818 | 0.000432958 | intron 1 of 5 | 1646 | Clic1 | 1.53017017 | 0.908403191 | |
| chr17-35049442-35049978 | 1.170057277 | 0.001863821 | 0.005497085 | Promoter | 0 | Clic1 | 1.53017017 | 0.903060701 | |
| chr12-113142362-113143009 | 2.20121083 | 2.78426E-08 | 4.50308E-07 | 5' UTR | 2126 | Crip2 | -1.179624675 | 0.874651945 | |
| chr1-165763426-165764536 | 0.698627193 | 0.001344608 | 0.004110077 | Promoter | 0 | Creg1 | 3.182864498 | 0.863872658 | |
| chr3-115714473-115715572 | 0.529473738 | 0.001090909 | 0.003419779 | Promoter | 0 | S1pr1 | 6.078338735 | 0.85868782 | |
| chr1-144003732-144004869 | 0.503180854 | 0.000973157 | 0.003093617 | Promoter | 0 | Rgs2 | -1.029459407 | 0.84860628 | |
| chr8-11259710-11260316 | 2.934023614 | 5.02565E-08 | 7.16949E-07 | intron 2 of 51 | 52510 | Col4a1 | 2.874398013 | 0.836726765 | 51 |
| chr2-13573367-13574831 | 0.982793248 | 1.80708E-06 | 1.34356E-05 | Promoter | 0 | Vim | 3.642359672 | 0.833198156 | 52 |
| chr6-5394843-5395471 | 3.417825897 | 3.2162E-07 | 3.25948E-06 | intron 2 of 5 | 11457 | Asb4 | 7.324761549 | 0.824485197 | 53 |
| chr6-97409384-97410387 | 4.228034629 | 1.05994E-12 | 2.01328E-10 | intron 5 of 23 | 207154 | Frmd4b | 1.665242286 | 0.79211103 | |
| chr8-94892676-94893416 | 5.286881825 | 1.17385E-11 | 6.55052E-09 | Distal Intergenic | -16346 | Dok4 | 4.236953185 | 0.791249714 | |
| chr4-115052371-115053185 | 1.449178182 | 2.1041E-06 | 1.52746E-05 | Distal Intergenic | -3241 | Tal1 | 7.96371719 | 0.783816443 | 54 |
| chr5-142920628-142921318 | 2.070287601 | 3.6157E-05 | 0.000170398 | Distal Intergenic | -13874 | Actb | 0.159559117 | 0.770046744 | |
| chrX-109012487-109014210 | 0.723851326 | 0.000378802 | 0.001349457 | Promoter | 0 | Hmgn5 | -2.106915277 | 0.76823996 | |
| chr11-73176388-73178047 | 0.727610704 | 3.78039E-05 | 0.00017684 | Promoter | 0 | Tax1bp3 | 1.672449365 | 0.740025424 | |
| chr2-93324461-93325216 | 2.686220682 | 1.77961E-06 | 1.32732E-05 | intron 1 of 9 | 9289 | Tspan18 | 4.721226361 | 0.727564885 | |
| chr1-135740911-135741635 | 1.589350233 | 0.000158184 | 0.000621694 | intron 2 of 5 | 20850 | Csrp1 | 2.379337052 | 0.724418472 | 55 |
| chr3-36475267-36476308 | 0.73214147 | 0.00016777 | 0.000654584 | Promoter | 0 | Anxa5 | 3.572949326 | 0.721486433 | |
| chr14-31403865-31404494 | 2.648559146 | 1.39687E-05 | 7.5035E-05 | intron 3 of 10 | 31584 | Sh3bp5 | 1.72053431 | 0.719007757 | |
| chr16-19946514-19947408 | 3.489457448 | 1.90217E-07 | 2.09978E-06 | 3' UTR | 35629 | Klhl6 | 7.90683331 | 0.691951993 | |
| chr2-26588502-26589118 | 1.947153026 | 4.0346E-06 | 2.63323E-05 | 5' UTR | 8488 | Egfl7 | 3.170311138 | 0.690434063 | 56,57 |
| chr3-115689326-115690014 | 1.917863191 | 0.000293159 | 0.001077617 | Distal Intergenic | 25058 | S1pr1 | 6.078338735 | 0.688486545 | |

**Table 2 (continued) | Chromatin regions opened at d4 and mapped to EC marker genes of a Tbx1Cre - selected EC cluster**

| Genome coordinates* | logFC (ATAC) | P value | FDR | Location | Distance To TSS | Gene | Log2FC (gene exp) | Prediction score | Reference |
|---|---|---|---|---|---|---|---|---|---|
| chr11-30160959-30161664 | 1.728503675 | 6.59343E-08 | 8.88461E-07 | intron 2 of 35 | 106511 | Sptbn1 | 0.92204993 | 0.681075617 | |
| chr6-97349325-97350358 | 3.267126204 | 8.94186E-12 | 9.74316E-10 | intron 8 of 23 | 267183 | Frmd4b | 1.665242286 | 0.672500964 | |
| chr5-75962059-75962768 | 3.525514843 | 6.44941E-09 | 1.42441E-07 | exon 10 of 30 | 15690 | Kdr* | 9.563929816 | 0.659810022 | 58 |
| chr7-25688933-25689749 | 1.080300981 | 0.000399915 | 0.001416685 | 5' UTR | 1931 | Tgfb1 | 3.815157444 | 0.657454308 | 59 |
| chr8-33853390-33854117 | 3.063001529 | 2.10081E-08 | 3.61268E-07 | exon 1 of 6 | 75746 | Rbpms | -0.220039076 | 0.653527968 | |
| chr5-142925880-142926955 | 1.078374232 | 3.27414E-05 | 0.000156548 | Distal Intergenic | -19126 | Actb | 0.159559117 | 0.647794343 | |
| chr8-84701265-84702036 | 1.101006127 | 0.000391951 | 0.001390223 | Promoter | 0 | Lyl1 | 6.327509169 | 0.647382071 | 60 |
| chr11-30166030-30166678 | 3.429683301 | 1.57673E-08 | 2.87993E-07 | intron 2 of 35 | 101497 | Sptbn1 | 0.92204993 | 0.641660249 | 61 |
| chr2-26474918-26475477 | 1.799531775 | 0.000513476 | 0.00176198 | intron 15 of 33 | 41186 | Notch1* | 0.257312471 | 0.62412906 | 57 |
| chr14-63957637-63958271 | 2.390048736 | 5.94194E-05 | 0.000261775 | Distal Intergenic | 13964 | Sox7 | 6.987177117 | 0.59807373 | 57 |
| chr3-145798563-145799322 | 4.210806179 | 7.79606E-09 | 1.65056E-07 | intron 1 of 5 | 39888 | Ddah1 | -0.720051753 | 0.585707542 | |
| chr12-73708585-73709289 | 2.265888997 | 1.68184E-05 | 8.80233E-05 | intron 10 of 14 | 123789 | Prkch | 1.45445198 | 0.581854654 | |
| chr2-85140743-85141322 | 5.553067474 | 3.86613E-10 | 1.61008E-08 | Distal Intergenic | 4518 | Aplnr | 9.097290391 | 0.574857801 | |
| chr3-93554391-93556403 | 2.214924157 | 4.05787E-13 | 1.05751E-10 | Promoter | 0 | S100a10 | 5.054205204 | 0.572158829 | |
| chr16-19982519-19983410 | 3.86140844 | 1.0683E-07 | 1.32511E-06 | Promoter | 0 | Klhl6 | 7.90683331 | 0.568797524 | |
| chr11-30157983-30158519 | 4.568785303 | 5.51597E-08 | 7.72084E-07 | intron 3 of 35 | 109656 | Sptbn1 | 0.92204993 | 0.553206105 | |
| chr6-129532510-129533841 | 0.789252522 | 0.000273125 | 0.001011214 | Promoter | 0 | Gabarapl1 | 0.372523642 | 0.525039134 | |
| chr1-173329069-173329822 | 4.05952659 | 1.29572E-14 | 1.05559E-11 | Distal Intergenic | 3928 | Ackr1 | 1.440784328 | 0.514067727 | |
| chr2-93353621-93354182 | 2.895760309 | 3.23524E-06 | 2.18322E-05 | Distal Intergenic | -19116 | Tspan18 | 4.721226361 | 0.510984458 | |
| chr3-109363425-109364021 | 3.396376381 | 9.57111E-07 | 7.93272E-06 | intron 1 of 26 | 22772 | Vav3 | 2.623446553 | 0.502212038 | |
| chr16-95441060-95441679 | 3.363920206 | 2.39211E-07 | 2.55148E-06 | intron 2 of 10 | 144914 | Erg | 6.889513443 | 0.499469476 | 62 |
| chr8-11233952-11234565 | 3.537800237 | 1.19408E-06 | 9.50415E-06 | intron 21 of 51 | 78261 | Col4a1 | 2.874398013 | 0.495299847 | |
| chr1-172501596-172502235 | 1.089221362 | 0.000165125 | 0.000645861 | intron 1 of 1 | 1549 | Tagln2 | 1.388094983 | 0.463370767 | |
| chr10-26799417-26800134 | 2.960095076 | 1.1126E-07 | 1.36664E-06 | intron 1 of 4 | 45996 | Arhgap18 | -0.870362617 | 0.456150298 | |
| chr1-64040678-64041514 | 1.686016644 | 1.70116E-05 | 8.88654E-05 | intron 3 of 3 | 80768 | Klf7 | -0.949199669 | 0.443756598 | |
| chr9-95316528-95317370 | 3.494600452 | 8.82422E-10 | 3.00225E-08 | Distal Intergenic | 89352 | Chst2 | 3.037623284 | 0.442088425 | |
| chr2-85131555-85132201 | 2.711506803 | 6.0688E-09 | 1.35764E-07 | Distal Intergenic | -4024 | Aplnr | 9.097290391 | 0.431132704 | 63 |
| chr18-53425671-53426379 | 2.794705195 | 7.74867E-08 | 1.01125E-06 | Distal Intergenic | -7556 | Ppic | 5.0736309 | 0.425441329 | |
| chr14-75138353-75138975 | 3.943462533 | 1.38032E-07 | 1.62314E-06 | intron 1 of 2 | 7252 | Lcp1 | 3.84734392 | 0.417696009 | 64 |
| chr18-53426927-53427686 | 4.870005269 | 1.10308E-12 | 2.06554E-10 | Distal Intergenic | -8812 | Ppic | 5.0736309 | 0.415547817 | |
| chr3-89835050-89835826 | 3.075642121 | 3.07665E-06 | 2.09353E-05 | intron 2 of 5 | 3680 | She | 3.765547855 | 0.412298789 | |
| chr10-4282309-4282866 | 2.362020174 | 3.2946E-05 | 0.000157318 | intron 1 of 3 | 15929 | Akap12 | -0.870447807 | 0.390930968 | |
| chr16-75908881-75909529 | 3.651681991 | 8.30462E-08 | 1.06999E-06 | 5' UTR | 112752 | Samsn1 | 5.666546058 | 0.365464515 | |
| chr1-194927690-194928341 | 2.724633467 | 2.77687E-07 | 2.884E-06 | Distal Intergenic | -10478 | Cd34 | 0.44042767 | 0.356837904 | |
| chr10-4305815-4306601 | 3.424551318 | 5.51123E-09 | 1.25966E-07 | intron 1 of 3 | 39435 | Akap12 | -0.870447807 | 0.355125106 | 65 |
| chr9-95380768-95381560 | 2.442319238 | 2.5521E-06 | 1.79756E-05 | Distal Intergenic | 25162 | Chst2 | 3.037623284 | 0.350824132 | |
| chr3-109378581-109379181 | 3.811986044 | 8.44848E-08 | 1.08576E-06 | intron 1 of 26 | 37928 | Vav3 | 2.623446553 | 0.339812462 | |

**Table 2 (continued) | Chromatin regions opened at d4 and mapped to EC marker genes of a Tbx1Cre - selected EC cluster**

| Genome coordinates* | logFC (ATAC) | P value | FDR | Location | Distance To TSS | Gene | Log2FC (gene exp) | Prediction score | Reference |
|---|---|---|---|---|---|---|---|---|---|
| chr13-60897290-60898110 | 3.410979205 | 1.42447E-07 | 1.66633E-06 | Promoter | 0 | Ctla2b | 2.673239312 | 0.335291515 | |
| chr9-52027082-52027944 | 3.001473815 | 4.88859E-08 | 7.00005E-07 | Distal Intergenic | -19229 | Rdx | 0.482386098 | 0.318892828 | |
| chr13-89401033-89401821 | 3.162063423 | 1.21102E-09 | 3.86993E-08 | Distal Intergenic | -137975 | Hapln1 | 8.244174051 | 0.31393253 | |
| chr5-147657739-147658344 | 3.679060144 | 3.59534E-08 | 5.50161E-07 | intron 10 of 29 | 67667 | Flt1* | 3.457437932 | 0.312128356 | |
| chr2-85123744-85124380 | 0.863115881 | 0.002825274 | 0.007924122 | Distal Intergenic | -11845 | Aplnr | 9.097290391 | 0.311506179 | |
| chr9-114808599-114809249 | 3.337893079 | 6.67984E-08 | 8.97712E-07 | intron 1 of 3 | 34907 | Cmtm8 | 1.106835823 | 0.30141788 3 | |
| chr2-84389089-84389687 | 4.385089165 | 1.18867E-07 | 1.44324E-06 | intron 2 of 15 | 35724 | Calcrl | 5.437845621 | 0.287387375 | |
| chr7-92883715-92884361 | 4.108801752 | 4.48011E-08 | 6.52378E-07 | intron 1 of 8 | 9245 | Prcp | 0.890127682 | 0.2870828 | |
| chr6-97582577-97583222 | 2.141686723 | 6.19429E-05 | 0.000271944 | intron 1 of 23 | 34319 | Frmd4b | 1.665242286 | 0.274362384 | |
| chr17-43373934-43374631 | 3.026626647 | 8.29606E-07 | 7.0874E-06 | intron 1 of 20 | 13483 | Adgrf5 | 4.69560476 | 0.269814575 | |
| chr16-76065857-76066646 | 3.693763761 | 1.16434E-08 | 2.26584E-07 | Distal Intergenic | -43576 | Samsn1 | 5.666546058 | 0.269170012 | |
| chr13-89335818-89336406 | 2.395069713 | 0.000641372 | 0.002142907 | Distal Intergenic | -203390 | Hapln1 | 8.244174051 | 0.26824179 7 | |
| chr9-95314673-95315374 | 2.884049845 | 1.15305E-06 | 9.24747E-06 | Distal Intergenic | 91348 | Chst2 | 3.037623284 | 0.264828778 | |
| chr13-89547208-89548072 | 3.499979281 | 5.14707E-10 | 1.98343E-08 | intron 1 of 4 | 7412 | Hapln1 | 8.244174051 | 0.262742598 | |
| chr13-89484275-89485097 | 2.772654798 | 1.03266E-05 | 5.79204E-05 | Distal Intergenic | -54699 | Hapln1 | 8.244174051 | 0.255978068 | |
| chr3-83382519-83383162 | 3.521621792 | 7.1222E-08 | 9.43617E-07 | Distal Intergenic | -120872 | Mef2c | 3.644161905 | 0.254208216 | |
| chr3-145813630-145814226 | 3.318682751 | 2.98568E-06 | 2.04041E-05 | intron 1 of 5 | 54955 | Ddah1 | -0.720051753 | 0.253784785 | |
| chr12-58367729-58368337 | 2.905737944 | 4.15522E-06 | 2.69126E-05 | Distal Intergenic | -98439 | Clec14a | not expressed | 0.249409401 | |
| chr3-109286346-109287236 | 5.452417009 | 6.17653E-16 | 1.76511E-12 | Distal Intergenic | -53417 | Vav3 | 2.623446553 | 0.240780479 | |
| chr13-13493413-13494152 | 3.674686103 | 7.48442E-09 | 1.60066E-07 | intron 12 of 19 | 55862 | Nid1 | 4.491278717 | 0.239005797 | |
| chr3-115659308-115659933 | 8.846757077 | 1.25958E-08 | 2.41402E-07 | Distal Intergenic | 55139 | S1pr1 | 6.078338735 | 0.235343788 | |
| chr3-89496485-89497058 | 4.010793625 | 2.69117E-07 | 2.81291E-06 | Distal Intergenic | -42738 | Hapln1 | 8.244174051 | 0.230621683 | |
| chr3-144707199-144707864 | 3.040975659 | 1.08953E-07 | 1.3421E-06 | intron 3 of 9 | 12471 | Sh3glb1 | 1.483562543 | 0.21845483 | |
| chr3-89493047-89493859 | 3.671341029 | 4.39692E-10 | 1.78125E-08 | Distal Intergenic | -45937 | Hapln1 | 8.244174051 | 0.218100386 | |
| chr13-106836013-106836912 | 2.462918282 | 1.00218E-07 | 1.25417E-06 | 3' UTR | 100046 | Ipo11 | -0.279800828 | 0.196332462 | |
| chrX-114473775-114474757 | 1.585786837 | 4.09451E-06 | 2.66422E-05 | Promoter | 0 | Klhl4 | | 0.185693143 | |
| chr10-77193427-77194055 | 1.478798032 | 0.002767985 | 0.007785036 | Distal Intergenic | -26879 | Col18a1 | 0.159193727 | 0.169242251 | |

* Genome coordinates shown in bold refer to regions with prediction score >0.5 (more than 50% chance to be enhancer).

**Table 3 | Putative EC regulatory regions associated with the genes indicated**

| Peak region | Location | Distance to TSS | Gene | Upregulated at d4 compared to d2 | Prediction score |
|---|---|---|---|---|---|
| chr2-32661523-32662398 | intron 2 | 14928 | *Eng* | yes | 0.7015349 |
| chr5-75961876-75963126 | intron 10 | 15332 | *Kdr* | yes | 0.6563047 |
| chr2-26474950-26475427 | intron 15 | 41236 | *Notch1* | no | 0.6256454 |
| chr8-104110355-104111309 | intron 1 | 8730 | *Cdh5* | yes | 0.5127832 |
| chr11-106713708-106714257 | intron 2 | 36371 | *Pecam1* | yes | 0.4803304 |
| chr5-147657600-147658539 | intron 10 | 67472 | *Flt1* | yes | 0.3063491 |

*TSS:* Transcription start site.

putative EC-REs identified had a score above 0.5, indicating a high probability of being enhancers (Table 3 and Fig. 4). Furthermore, the two putative EC-REs associated with *Kdr* and *Notch1* are amongst the 101 regions open at d4 and mapped to EC marker genes (asterisks on Table 2).

To test the importance of the putative EC-REs, we used an epigenetic reprogramming strategy based on CRISPR-dCAS9:LSD1 (Fig. 5a). We first generated an ES cell line that stably expressed the dCAS9:LSD1 construct (named #B1 dCas9-LSD1). We then designed crRNAs targeting the six EC-REs and a control crRNA targeting a gene desert sequence (Supplementary Data 5). We transfected #B1 dCas9-LSD1 cells with targeting and control gRNAs complex (crRNAs:ATTO-tagged tracrRNA), FACS-purified the ATTO+ cells, and subjected them to the EC differentiation protocol. Cells were harvested at d4, d6, and d8 and the expression of the targeted genes was measured by quantitative real-time PCR (qPCR). Experiments were repeated at least four times. Results showed that LSD1 targeting of the six EC-REs resulted in reduced expression of the associated genes (Fig. 5b), with the exception of the *Pecam1*-associated RE, which also had a low probability score (Table 3). In most cases, the reduction in gene expression was more evident at the later stages of differentiation tested, namely d6 and, even more so at d8.

### The EC-REs for Notch1 and Pecam1 are required for gene expression

Next, we selected the *Notch1* and *Pecam1* EC-REs for further testing based on the importance of the associated genes for EC differentiation, and because of the negative results obtained with epigenetic reprogramming for *Pecam1*. We deleted the putative EC-REs using CRISPR-Cas9; for each RE, we selected two gRNAs flanking the segment (Fig. 6a, gRNA sequences are shown in Supplementary Data 5), and these were transfected into mESCs along with the Cas9 protein and ATTO-labeled tracrRNA. We then plated FACS-purified ATTO+ cells to a clonal density. Clones were later picked and expanded into 96-well plates. DNA extracted from clones was screened by PCR to identify clones carrying homozygous deletion of the putative RE. We expanded two homozygously deleted clones for each deleted RE. All four clones were used in multiple differentiation experiments (*n* = 5). Results showed that both *Pecam1* and *Notch1* expression were significantly affected by the deletion of their respective REs at d6 and d8 (Fig. 6b).

We next tested whether the deletion of the *Notch1* EC-RE affected the expression of a subset of NOTCH1 target genes, namely *Hes1, Nrarp*, and *Dll4*. Results showed that all of these genes were affected by the deletion of the enhancer, but with some differences. Specifically, *Hes1* and *Nrarp* were significantly and consistently downregulated at d6 but not at d8. Conversely, *Dll4* was downregulated only at d8 (Fig. 6c). Thus, deletion of the EC-RE identified here was sufficient to cause dysregulation of at least part of the NOTCH1 signaling pathway.

Next, we used the *Notch1* enhancer deletion lines (clones #7 G, #11B *Notch1*-Δenh.in15), along with the parental WT line, to generate gastruloids as described by ref. [23]. Gastruloids developed a primitive EC network (PECAM1-positive) in both the WT and mutant lines (examples in Fig. 7a, *n* = 10 immunostained for each clone). However, mutant gastruloids appeared more densely stained, although we could not quantify them due to the complexity of patterns. Therefore, we subjected the lines

to a standard Matrigel test and evaluated the branching points of the EC network generated (see Methods). Results of five independent experiments showed that *Notch1* mutant clones developed a more intricated EC network with a significantly higher number of branch points (Fig. 7b). These results are consistent with the NOTCH1 signaling role in limiting vessel branching[24].

### Discussion

The cardiopharyngeal mesoderm[6] is a lineage that provides progenitors to various structures including those of the heart, pharyngeal apparatus, and vessels[9,10]. Endothelial cells are heterogeneous in origin and single-cell sequencing assays are starting to define specific transcription and chromatin profiles depending on the tissue of origin[4]. However, whether cells destined to differentiate in EC are primed by distinct mechanisms according to their origin, is still unclear. One possible avenue to address this question is to identify tissue-specific enhancers for each lineage. In this study, we propose an approach that leverages novel and published data, integrated with software tools, and genetic/epigenetic editing in a cell differentiation model. This integrated approach identified a group of putative EC enhancers, some of which had already been reported in the literature and were validated in our model. For this study, we used a mesoderm differentiation protocol originally proposed for cardiogenic mesoderm induction[17] and observed the activation of EC-specific gene expression and chromatin remodeling (as assayed by ATAC-seq) 48 h after induction. However, at this stage (d4), only a small percentage of cells exhibit an EC phenotype as defined by the expression of VE-cadherin on the cell surface, suggesting that activation of the EC program is at an early stage. Boosting VEGF signaling after mesoderm induction promoted EC differentiation such that a near-homogeneous EC-like population was obtained at d8, as measured by VE-cadherin expression and Matrigel assays. We selected to leverage chromatin dynamics of EC-specific gene activation at an early developmental time window (d2-d4) in order to capture the regulatory sequences associated with the activation of the EC program in cardiogenic mesoderm. To this end, we used data-rich bulk ATAC-seq and RNA-seq information, combined with published high-resolution tissue- and time-specific scRNA-seq of cells that were selected using the *Tbx1^cre^* driver, *Tbx1* being a marker of the cardiopharyngeal mesoderm. This enabled us to identify 101 regions that became more accessible in the selected time window and mapped to EC genes defined by scRNA-seq data (Table 2). Of the 57 putative enhancers scoring >0.5 identified through our unbiased approach, 15 (26%) were already reported in the literature, thus suggesting that our approach was efficient in detecting likely regulatory elements in our model. In addition, motif analysis showed enrichment of transcription factors known to be involved in EC development, further supporting the suitability of the approach. However, we did not validate these putative enhancers directly in our model; thus, further work will be necessary to establish the reliability of our approach for systematic identification of cell type-specific enhancer sequences.

The candidate gene approach was designed to identify regulatory sequences "activated" in our model and associated with genes known to be involved in EC development. We have validated them regardless of the prediction score and found five of the six tested putative REs to regulate the respective genes. Overall, we identified regulatory elements for many of

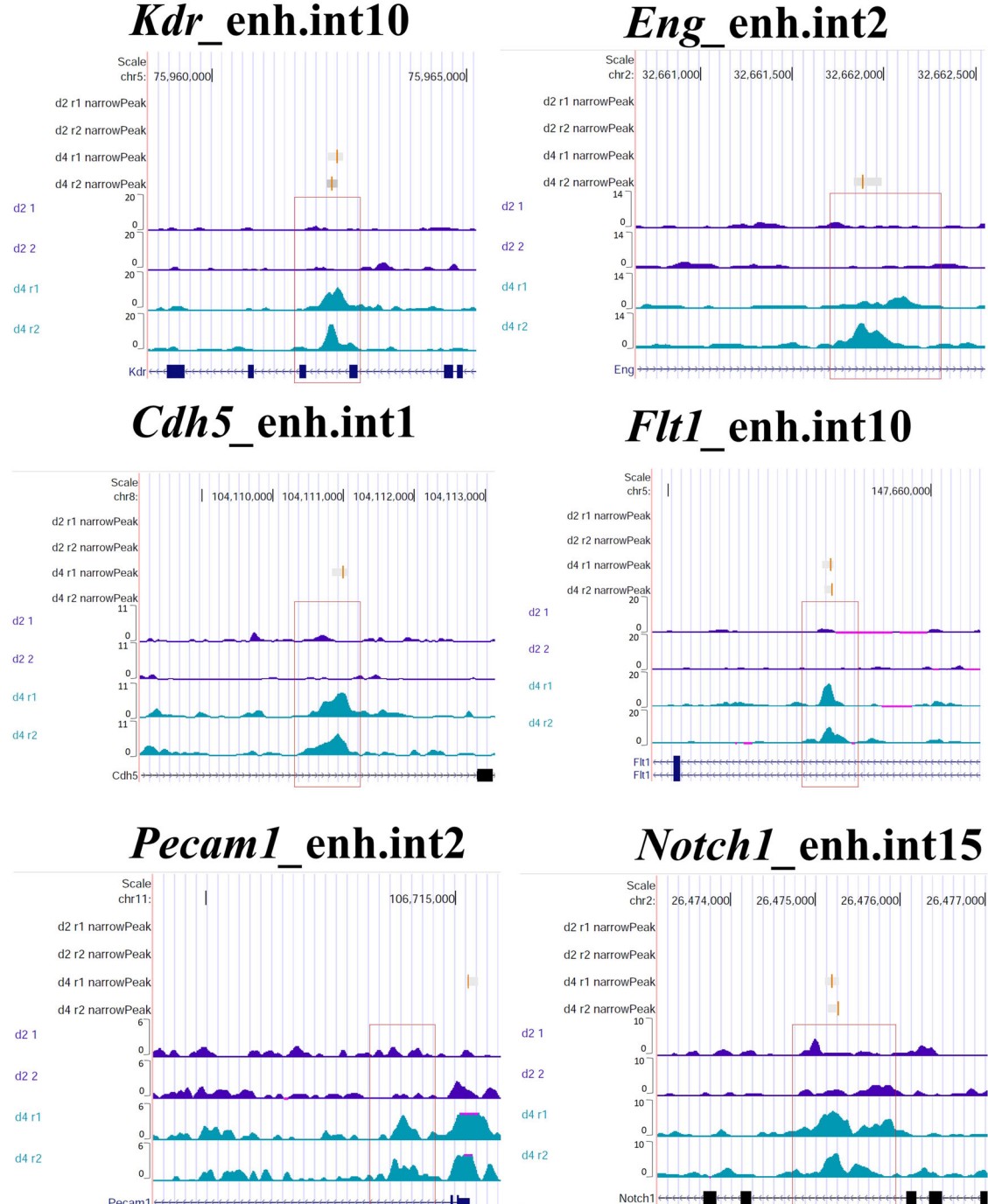

**Fig. 4 | Selection of putative regulatory elements in major EC genes (EC-REs).** ATAC-seq coverage associated with six putative EC-REs, associated with six selected EC genes: *Kdr; Eng; Cdh5; Flt1; Pecam1; Notch1.* On the vertical axis, there are the genome coverage of d2 (replicate1 and replicate2) and d4 (replicate1 and replicate2). Red boxes indicate the open chromatin region at d4.

the known genes involved in EC development, including a subset of genes expressed in CPM-derived cells in vivo like, for example, *Notch1*[8].

Epigenetic reprogramming, while providing consistent results, proved to be variable in our hands. Sources of variability may be the efficiency of transfection, the gRNAs, or perhaps the variable extent of chromatin modification induced by the dCAS9:LSD1 complex. Furthermore, the inconsistent results obtained with the *Pecam1* putative enhancer using epigenetic reprogramming and gene editing may be due to different reasons. We speculate that perhaps the sequence is not a regulatory element (as suggested by the low prediction score), but the genetic deletion may have

altered the expression of the gene by interfering with processes like RNA maturation/splicing or causing other structural perturbation of the gene.

The dCas9-recruited repressor could potentially cause chromatin modifications beyond the intended targeted sequence, particularly if the promoter is nearby. The six enhancers tested with this method are all fairly distant from the transcriptions start site (TSS, Table 3). The closest is 8.7Kb from the TSS, the others are between 15 and 67 Kb.

Searches for consensus sequences in the putative enhancer regions identified GATA motifs as the most enriched, together with ETS factors (ERG, FLI,) and the AP-1 subunit FOS (Fig. 3b). EC genes from cell clusters

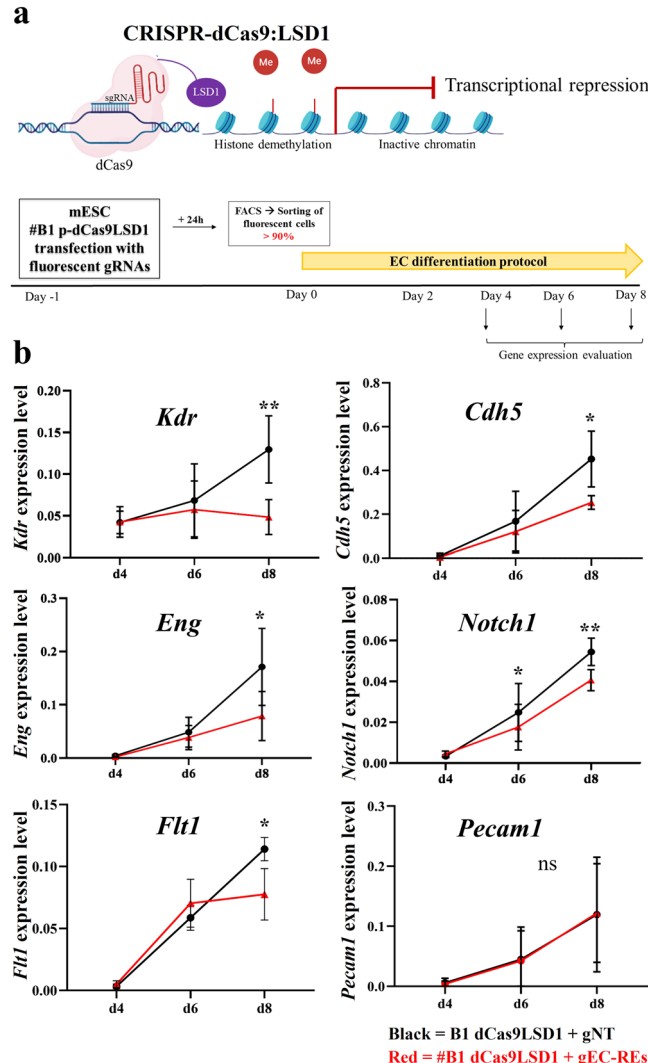

**Fig. 5 | Epigenetic reprogramming validates putative regulatory elements. a** Top: Schematic overview of CRISPR-dCas9:LSD1 system: the fusion protein dCas9:LSD1 is able to bind DNA target and LSD1 can demethylate histone H3 lysine 4 (H3K4me1 and me2) near the putative enhancer region to decommission the enhancer. *Bottom*: cartoon of the experimental plan. mESC #B1 dCas9-LSD1 were transfected with fluorescent gRNAs. Fluorescent-sorted cells were differentiated into ECs from day 0 to day 8. Samples were collected on day 4, day 6, and day 8 to analyze the gene expression. **b** Quantitative real-time PCR (qPCR) analysis of *Kdr; Cdh5; Eng; Notch1; Flt1*, and *Pecam1* mRNA expression level in cells of clone #B1 dCas9-LSD1 transfected with gRNAs targeted (in red) or control (in black) during EC differentiation. X-axis denotes the three time points (d4-d6-d8); y-axis indicates the expression level, evaluated using the $2^{-\Delta Ct}$ method. *Gapdh* expression was used as a normalizer. Values are the average of four ($n = 4$) biological replicates $\pm$ standard deviation (SD). $p$ value (*) <0.05 and $p$ value (**) <0.01 are considered significant; ns no statistical significance (parametric paired *t*-test, one-tailed).

derived using the *Mesp1^Cre* driver[22], which captures a larger and more diverse EC population than *Tbx1^Cre*, exhibited DARs enriched for a more extensive set of motifs, but they also included GATA and ETS motifs. Interestingly, the *Mesp1^cre* dataset motifs also included transcription factor families that play a role in CPM development, such as T-BOX, FOX, and MEIS factors (Supplementary Data 4), raising the question of whether they may be involved in enabling the EC transcription program in the CPM.

Overall, the results of consensus sequence searches suggest that there is a core of transcription factors, GATA-ETS-FOS, that are central to the activation of the EC program in our model. There is ample literature indicating GATA and ETS transcription factors as general players in endothelial differentiation (reviewed in De Val and Black, 2009)[2]. It is

therefore possible that during mesoderm induction in our system, GATA factors act as pioneers to establish the conditions necessary for the binding of other, lineage determining factors, such as ETS/ERG. This is consistent with the established role of GATA factors as pioneer transcription factors (review in refs. [25,26]) and the role of ETS factors as core transcription factors in endothelial differentiation[3,27–31].

Overall, our strategy efficiently identified putative enhancers of cell type-specific genes during differentiation. It provided us with an extensive list of regulatory sequences with probability scores calculated using a machine-learning approach. Furthermore, it allowed us to identify and validate a smaller set of regulatory sequences of well-known genes involved in EC differentiation. The identification and validation strategies applied here are applicable to other cell types, whenever a suitable differentiation model is available, although more extensive bench validation experiments will be required before the proposed approach may be considered an established pipeline for enhancer identification.

## Methods
### Mouse embryonic stem cells (mESC) culture and manipulation
ES-E14TG2a mESCs (ATCC CRL-1821) were cultured without feeders and maintained undifferentiated on gelatin-coated dishes in GMEM (Sigma Cat# G5154) supplemented with $10^3$ U/ml ESGRO LIF (Millipore, Cat# ESG1107), 15% fetal bovine serum (ES Screened Fetal Bovine Serum, US Euroclone Cat# CHA30070L), 0.1 mM non-essential amino acids (Gibco, Cat# 11140-035), 0.1 mM 2-mercaptoethanol (Gibco, Cat# 31350-010), 0.1 mM L-glutamine (Gibco, Cat# 25030081), 0.1 mM Penicillin/Strepto-mycin (Gibco, Cat# 10378016), and 0.1 mM sodium pyruvate (Gibco, Cat# 11360-070). Cells were passaged every 2–3 days using 0.25% Trypsin-EDTA (1X) (Gibco, Cat# 25200056) as the dissociation buffer.

For differentiation, E14-Tg2a mESCs were dissociated with Trypsin-EDTA and cultured at 75,000 cells/ml in serum-free media: 75% Iscove's modified Dulbecco's media (Cellgro Cat# 15-016-CV) and 25% HAM F12 media (Cellgro #10-080-CV), supplemented with N2 (GIBCO #17502048) and B27 (GIBCO #12587010) supplements, penicillin/streptomycin (GIBCO #10378016), 0.05% BSA (Invitrogen Cat#. P2489), L-glutamine (GIBCO #25030081), 5 mg/ml ascorbic acid (Sigma A4544) and $4.5 \times 10^{-4}$ M monothioglycerol (Sigma M-6145). After 48 h in culture, the EBs were dissociated using the Embryoid Body dissociation kit (cod. 130-096-348 Miltenyi Biotec) according to the manufacturer's protocol and reaggregated for 40 h in serum-free differentiation media with the addition of 8 ng/ml human Activin A (R&D Systems Cat#. 338-AC), 0.5 ng/ml human BMP4 (R&D Systems Cat# 314-BP), and 5 ng/ml human-VEGF (R&D Systems Cat#. 293-VE). The 2-day-old EBs were dissociated and $6 \times 10^4$ cells were seeded onto individual wells of a 24-well plate coated with 0.1% gelatin in EC Induction Medium consisting of StemPro-34 medium (Gibco #10639011), supplemented with SP34 supplement, L-glutamine, penicillin/streptomycin, 200 ng/ml human-VEGF, and 2 µM Forskolin (Abcam, ab120058). The Induction Medium was changed after 1 day. On day 6 of differentiation, the cells were dissociated and replated on 0.1% gelatine-coated dishes at a density of 25,000 cells/cm$^2$ in EC Expansion Medium, consisting of StemPro-34 supplemented with 50 ng/ml human-VEGF. Stem cell-derived endothelial cells were maintained until they reached confluency (about 2–3 days). EC Expansion Medium was replaced every other day.

For the Matrigel assay, 300 µL of Matrigel (BD Matrigel Basement Membrane Matrix Growth Factor Reduced, Phenol Red Free cat. 356231) was aliquoted into each well of a 24-well plate and incubated for 30–60 min at 37 °C to allow the gel to solidify. About 200,000 ECs were then added to the Matrigel-coated well and cultured for 24 h at 37 °C. Formation of tubular structures on a two-dimensional Matrigel surface was observed after 16 to 24 h under an optical microscope. The quantification of branch points was performed using the Angiogenesis Analyzer module[32] of Image J. The total number of branch points per image was quantified. We performed statistical analyses of branch counts using the parametric paired *t*-test, one-tailed.

**Fig. 6 | Homozygous deletion of putative regulatory elements of Notch1 and Pecam1 genes reduced their expression during EC differentiation. a** Scheme of the steps of targeted Pecam1-enh.int2 and Notch1- enh.int15 deletion with CRISPR/Cas9. Red lines indicate the position of the two gRNAs used. **b** Quantitative real-time PCR (qPCR) analysis of *Notch1* mRNA expression level in mESC *Notch1-*Δenh.in15 (clones #7 G; #11B) and *Pecam1* in mESC *Pecam1*Δenh.in2 (clones #7G; 5G) during EC differentiation. *Notch1* and *Pecam1* expression was reduced in mutant cell lines (in red), compared to WT cells (in black), used as control. The X-axis denotes the three time points (d4-d6-d8); the y-axis indicates the expression level, evaluated using the $2^{-\Delta Ct}$ method. *Gapdh* expression is used as a normalizer. Values are the average of five biological replicates ± standard deviation (SD). *p* value (*) <0.05; *p* value (**) <0.01 and *p* value (***) <0.001 are considered significant; ns no statistical significance (parametric paired *t*-test, one-tailed). **c** Quantitative real-time PCR (qPCR) analysis of gene expression of *Notch1*-related genes in mESC *Notch1-*Δenh.in15 (clones #7G; #11B) during EC differentiation. *p* value (*) <0.05 and *p* value (**) <0.01 are considered significant; ns no statistical significance (parametric paired *t*-test, one-tailed).

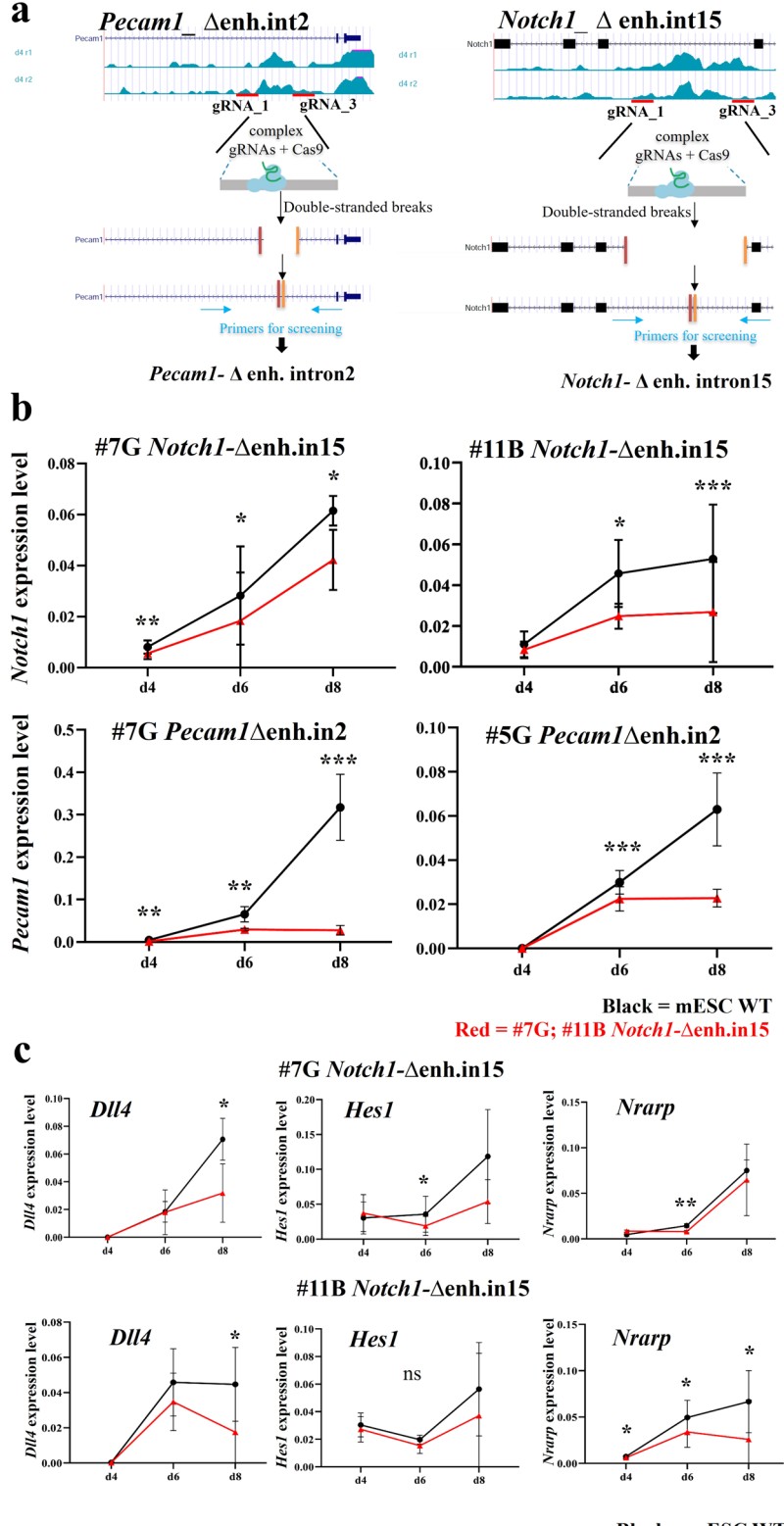

## Gastruloid formation assay

Gastruloids were generated as described in ref. [23]. In brief, 300 mESCs were plated in 40 μL N2B27 medium in 96-well Ultra-Low Cluster Round Bottom Ultra-Low Attachment plates (7007, Corning). After 48 h, 150 μL of N2B27 supplemented with 3 μM CHIR-99021 (S1263, Selleckchem) were added to each well. Then after 72 h, the medium was changed with 150 μL of fresh N2B27. At 96 h, gastruloids were transferred 1:1 in 100 μL of medium in 24-well Flat Bottom Ultra-Low Attachment plates (3473, Corning), containing 700 μL of fresh N2B27 supplemented with 30 ng/mL bFGF (Recombinant Human FGF-basic, 100-18 C, Peprotech), 5 ng/ml VEGF (Recombinant Human VEGF165, 100-20, Peprotech), and 0.5 mM ascorbic acid (Sigma A4544) (N2B27 +++). Then, 50% of the medium was changed daily, at 120 h with 400 μL of fresh N2B27 +++ and at 144 h with N2B27, until 168 h.

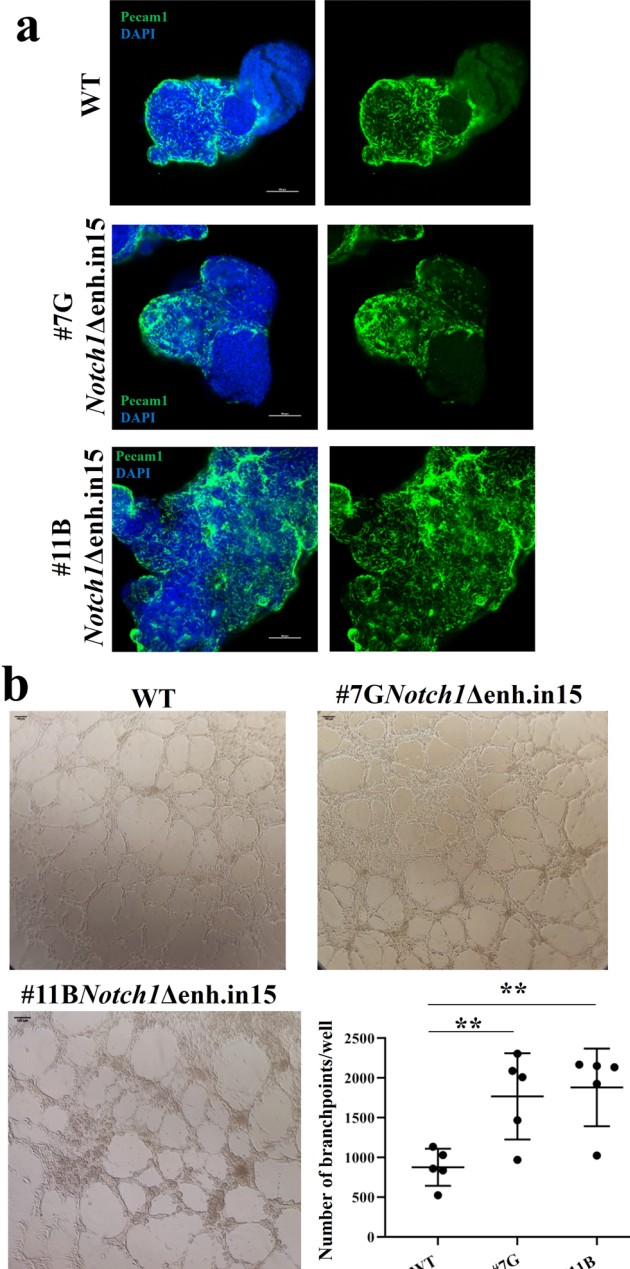

**Fig. 7 | The deletion of the Notch1 EC-RE affected the development of a vascular-like network in differentiating mESC. a** Immunofluorescence images showing PECAM1 expression (in green) on cardiac gastruloids at 168 h, using WT and mESC *Notch1*-Δenh.in15 (clones #7G; #11B). Images were obtained with Nikon A1 Confocal Microscopy. Scale bars: 100 μm. **b** In vitro tube formation assay (Matrigel) of d8 WT and mESC *Notch1*-Δenh.in15 (clones #7G; #11B) plated for 24 h. Scale bars, 100 μm. Quantification of branch points from five independent experiments. *p* value (**) <0.01 obtained using the parametric paired *t*-test, one-tailed. Error bars: SD.

## Immunofluorescence and confocal imaging on fixed gastruloids

For whole-mount immunofluorescence, gastruloids at 168 h were washed in 1x PBS and fixed in 4% PFA overnight at 4 °C while shaking. Then, fixed samples were washed three times in 1x PBS (5 min each) and three times (5 min each) in blocking solution1 (1x PBS, 10% Goat Serum, 0.1% Triton X-100) at 4 °C with agitation. Gastruloids were blocked for 1–2 h at 4 °C in blocking solution1 and then were incubated o.n. with primary antibody anti PECAM1 (mouse MA3105, Thermo Fisher Scientific) 1:200 in blocking solution1 at 4 °C with agitation. The day after, samples were washed two times (5 min each); then three times (15 min

each), and finally four to six times (for 1 h total) in blocking solution1 at 4 °C while shaking. Gastruloids were incubated o.n. with secondary antibody Goat Anti-Armenian hamster IgG H&L (Alexa Fluor® 488, Abcam ab173003) and DAPI in blocking solution1 at 4 °C while shaking. The day after, samples were washed two times (5 min each) in blocking solution1 at 4 °C; then two times (5 min each) at room temperature (RT) with blocking solution 2 (1x PBS, 0.2% Goat Serum, 0.2% Triton X-100) and finally three times (15 min each) in blocking solution 2 at RT while shaking. Subsequently, gastruloids were incubated in blocking solution 2/100% glycerol (Sigma) 1:1 for 30 min at RT with agitation. Then, they were maintained in plates with blocking solution 2/70% glycerol 7:3 at 4 °C. Images were acquired with Nikon A1 Confocal Microscopy (equipped with Nikon Resonant Scanner and NIS-A/NIS-Elements software).

## CRISPR-Cas9-mediated targeting

(A) *Pecam1* intron 2-enhancer deletion was induced in E14-Tg2a using Alt-R™ CRISPR-Cas9 System (IDT) following the manufacturer's specifications. This genome editing system is based on the use of a ribonucleoprotein (RNP) consisting of *S. pyogenes* Cas9 nuclease complexed with guide RNA (crRNA:tracrRNA duplex). The crRNA is a 20 nt custom-synthesized sequence that is specific for the target and contains a 16 nt sequence that is complementary to the tracrRNA. The specific crRNA sequences were: Pecam1_int2-crRNA1 and Pecam_int2-crRNA3 (sequences shown in Supplementary Data 5). CRISPR-Cas9 tracrRNA-ATTO 550 (5 nmol catalog no. 1075927) is a conserved 67 nt RNA sequence that is required for complexing to the crRNA so as to form the guide RNA that is recognized by S.p. Cas9 (Alt-R S.p. Cas9 Nuclease 3NLS, 100 μg catalog no. 1081058). The fluorescently labeled tracrRNA with ATTO™ 550 fluorescent dye is used to FACS-purify transfected cells. The protocol involves three steps: (1) annealing of the crRNA and tracrRNA, (2) assembly of the Cas9 protein with the annealed crRNA and tracrRNAs, and (3) delivery of the ribonucleoprotein (RNP) complex into mESC by reverse transfection. Briefly, we annealed equimolar amounts of resuspended crRNA and tracrRNA to a final concentration (duplex) of 1 μM by heating at 95 °C for 5 min and then cooling to room temperature. The RNA duplexes were then complexed with Alt-R S.p. Cas9 enzyme in OptiMEM media to form the RNP complex, which was then transfected into mESCs using the RNAiMAX transfection reagent (Invitrogen #13778-150). After 48 h incubation, cells were trypsinized and ATTO 550+, transfected cells were purified by FACS. Fluorescent cells (~65% of the total cell population) were plated at very low density to facilitate colony picking. We picked and screened PCR 96 clones. Primer sequences are indicated in the Supplementary Data 5. Positive clones were confirmed by DNA sequencing.

(B) For the *Notch1* intron 15-enhancer deletion, we followed the same procedure but using different target sequences: Notch1_int15-crRNA1 and Notch1_int15-crRNA3 (sequences shown in Supplementary Data 5).

## Generation of dCas9-LSD1 expressing mESC line

About 20 μg of plasmid p-dCas9-LSD1-Hygro (a gift from Stephan Beck and Anna Koeferle, available through Addgene plasmid #104406; http://n2t.net/addgene:104406; RRID:Addgene_104406) was linearized with *Ahd*I enzyme and electroporated in mESC (1 × 10⁷ cells/10 cm plate). The electroporation parameters used were 0.24 kV and 500 μF. The cells were maintained in Hygromicin B selection (500 μg/ml) for 10 days. Individual colonies were isolated, expanded, and screened by PCR for inserted sequences for both DNA and RNA. Primer sequences are in the Supplementary Data 5.

## CRISPR-dCas9:LSD1-mediated epigenetic reprogramming strategy

Epigenetic targeting of putative enhancer elements was induced by transfection of dCas9-LSD1-expressing mESC line with specific gRNA complex (crRNA:tracrRNA duplex). For each enhancer element, we designed three crRNA sequences (shown in Supplementary Data 5).

Then, we annealed equimolar amounts of resuspended crRNA and tracrRNA labeled with ATTO™ 550 fluorescent dye to a final concentration (duplex) of 1 µM by heating at 95 °C for 5 min and then cooling to room temperature. For gRNAtransfection, cells were plated at $8 \times 10^5$ per well in six-well plates and transfected with gRNA complex (crRNA:tracrRNA 10 nM) in an antibiotic-free medium using Lipofectamine RNAiMAX Reagent (Invitrogen #13778 - 150), according to the instructions of the manufacturer. Twenty-four hours after transfection, fluorescent ATTO 550+ cells (~90 – 95% of the total cell population) were harvested and subjected to the differentiation protocol. crRNA sequences are listed in the Supplementary Data 5.

## Flow cytometry
We dissociated cells with Trypsin-EDTA or with the Embryoid Body dissociation kit (cod. 130-096-348 Miltenyi Biotec). Dissociated cells ($1 \times 10^6$ cells/100 µl) were incubated with primary antibodies (VE-Cadherin-APC, mouse cod.130-102-738) directly conjugated (1:10) in PBS-BE solution (PBS, 0.5% BSA, 5 mM EDTA) for 20 min on ice. Subsequently, cells were washed twice with 2 ml of PBS-BE. Cells were analyzed using the BD FACS ARIAIII™ cell sorter. Negative controls were incubated with fluorochrome-labeled irrelevant isotype control antibody (REA Control APC, mouse cod. 130-113-446 Miltenyi Biotec).

## Quantitative RT-PCR
Total RNA was isolated from mouse ESCs with QIAzol lysis reagent (Qiagen #79306), according to the manufacturer's protocol. The isolated RNAs were quantified using a NanoDrop spectrophotometer 1000. Before reverse transcription, RNA samples were treated with DNAse I to eliminate any contamination with genomic DNA.

cDNA was transcribed using 1 or 2 µg total RNA with the High-Capacity cDNA reverse transcription kit (Applied Biosystem catalog. n. 4368814). cDNAs were amplified using myTaq™ DNA polymerase (Meridian Bioscience) and a standard three-step cycling PCR profile: 10 min at 94 °C, 30 amplification cycles (denaturation at 94 °C for 30 s, annealing at 60 °C for 30 s, and extension at 72 °C for 30 s), followed by a final extension at 72 °C for 10 min. Quantitative gene expression analyses (qRT-PCR) were performed using PowerUp™ SYBR Green Master Mix (Applied Biosystem #A25742). Relative gene expression was evaluated using the "$2^{-\Delta Ct}$" method, and *Gapdh* expression as a normalizer. cDNA was amplified by qRT-PCR, using StepOnePlus™ Real-Time PCR System. The run used was holding stage (95 °C - 10 min); cycling stage (95 °C – 15 s, 60 °C – 1 min for 40 cycles); melt curve stage (95 °C – 15 s, 60 °C – 1 min, 95 °C – 15 s). The cycle threshold (Ct) was determined during the geometric phase of the PCR amplification plots, as illustrated in the manufacturer's protocol. Expression data are shown as the mean ± SD. Primer sequences are listed in Supplementary Data 5. GraphPad Prism software v8.00 (GraphPad) was used to analyze qRT-PCR data. Relative mRNA levels were analysed in triplicate and data were presented as means ± SD. Two-way repeated measures ANOVA test (ANOVA two-way-RM) was used to assess the statistically significant interaction effect between "time" and "genotype" on a gene expression variable. Other two statistical methods between groups of data were used: nonparametric and parametric test. The first was a nonparametric Wilcoxon matched-pairs signed-rank test, one-tailed; the second statistical analysis were performed using the parametric Student's paired *t*-test, one-tailed. Shapiro–Wilk test was performed to determine the normality distribution of the dataset.

## RNA-seq
Total RNA was isolated from d2 (n. 2 biological replicates) and d4 (n. 2 biological replicates) cells with QIAzol lysis reagent (Qiagen #79306), according to the manufacturer's protocol. RNA concentration was estimated using a Nanodrop spectrophotometer 1000. Libraries were prepared according to the Illumina strand-specific RNA-seq protocol. Libraries were sequenced on the Illumina platform NextSeq500, in paired-end, 75 bp reads.

## ATAC-seq
mESCs were collected on day 2 and day 4 and then washed two times in PBS, harvested, counted using a hemacytometer chamber, and pelleted. About 15,000 cells/sample for mESC were treated with Tagment DNA Buffer 2x reaction buffer with Tagment DNA Enzyme (Illumina) according to the manufacturer's protocol. After washes in PBS, cells were suspended in 50 mL of cold lysis buffer (10 mM Tris-HCl, pH 7.4, 10 mM NaCl, 3 mM MgCl$_2$, 0.1% IGEPAL CA-630) and immediately spun down at 500 × g for 10 min at 4 °C. Fresh nuclei were treated with Transposition mix and Purification (Illumina #FC121-130), the nuclei were incubated at 37 °C in Transposition Reaction Mix (25 µL reaction buffer, 2.5 µL Transposase, 22.5 µL Nuclease-free water), purified using Qiagen MinElute PCR Purification Kit (catalog no./ID: 28006) and eluted in 10 µL of nuclease-free water. The sequencing library was prepared from the fragmented amplified tagmented DNA. Fragmentation size was evaluated using the Agilent 4200 TapeStation. Two biological replicates for each condition were sequenced using the Illumina NextSeq500 system to obtain paired-end (PE) reads of 60 bp.

## Sequence data analysis
For RNA-seq sequencing data, we assessed the quality of the paired-end (PE) reads of length 75 bp using FastQC. We filtered the low-quality and short PE reads and trimmed the universal Illumina adapters using cutadapt (v2.9)[33] by setting the following parameters: -q 30 -m 30. Post-trimming, we re-assessed the quality and compiled the report using multiQC. We aligned the PE reads to mm10/GRCm38 reference genome (primary assembly) using STAR aligner (v2.6.0a)[34] following two steps: (i) Generation of Ensembl mm10 reference genome index (release 102) setting the parameters --sjdbGTFfile 100 --sjdbOverhang 100 (ii) Alignment of PE reads to the reference genome (--sjdbOverhang 100 --quantMode GeneCounts --outSAMtype BAM SortedByCoordinate). We provided the sorted BAM as input and the mm10 (GRCm38.p4) primary assembly annotation file in GTF format(https://www.gencodegenes.org/mouse/release_M10.html) as reference annotation to quantify the gene expression levels with the featureCounts function from the Rsubread package (v2.0.1)[35] (annot.ext = " mm10.v102" gtf file, useMetaFeatures=TRUE, allowMultiOverlap=FALSE, strandSpecific=2, CountMultiMappingReads=FALSE). After that, we retained the expressed gene matrix by filtering out from the read count matrix the zero and low count genes (CPM <0.5) using the proportion test method from the NOIseq package (v2.34.0)[36]. Then, we processed the expressed gene matrix using NOIseq (v2.34.0), obtaining a set of normalized read counts (Upper Quartile, UQUA) and identified differentially expressed genes (DE) using the noiseq function and setting the posterior probability value cutoff >0.8. Simultaneously, we normalized the expressed count matrix with the sizefactor function in DEseq2[37] and we used the DESeq function with default options, then we selected the DE genes by setting the adj.*p* value cutoff <0.01. Finally, we considered DE genes in our study, those genes that were declared DEs from both methods. We performed gene ontology enrichment analysis using the gProfiler2 R package (v0.2.2)[38], providing the common DE gene list as input, the expressed gene list as background, and setting Benjamini–Hochberg FDR (BH-FDR) cutoff to <0.01.

ATAC-seq sequences underwent quality control (using FastQC and multiQC), adapter trimming, and filtering using cutadapt (v2.9) with parameters -q 30 -m 30 and universal Illumina adapters. Then, we aligned the PE sequences to the mouse genome (mm10) containing only canonical chromosomes with Bowtie2 (v2.3.4.3)[39], setting the options -q -t --end-to-end --very-sensitive -X 1000. After removing reads mapping to the mitochondrial chromosome, we removed duplicates and reads mapping to multiple positions using sambamba (v0.6.8) with -F "[XS] == null and not unmapped and not duplicate"[40]. We called the ATAC peaks for each sample using MACS2 (v2.1)[41] with the option -BAMPE –nomodel –shif100 –extsize 200, which are the suggested parameters to handle the Tn5 transposase cut site. After that, we removed those peaks overlapping the mm10-blacklist regions (downloaded from https://github.com/Boyle-Lab/Blacklist/blob/

master/lists/mm10-blacklist.v2.bed.gz) using samtools[42]. Then, for each condition, we defined the consensus lists of enriched regions as the peak regions common to both replicates using the intersectBed function from the BedTools v2.26.0[43]. To identify differentially enriched regions (DARs) between d4 and d2, we selected the regions consistently enriched or decreased in DEScan2 (v1.18.2)[20] and confirmed with the same sign also in DiffBind (v3.8.4)[21]. For DEScan2, we first created a peak-set using the finalRegions function (zThreshold = 1, minCarriers = 2) after loading all of the MACS2 peaks not overlapping blacklist regions. Then, we called the DARS with the edgeR[44,45] method as follows: we estimated the count dispersion using the estimateDisp function, we fitted the robust glmQLFit model and used the glmQLFTest, with the default parameters, setting the adj. $p$ value to 0.01. For DiffBind (v3.8.4), we set dba.count(minOverlap = 2), dba.contrast(minMembers = 2), dba.analyze(method = DBA_EDGER) and 0.01 as adj. $p$ value. Finally, we selected the DARs regions of DEScan2 confirmed by DiffBind using subsetByOverlaps in GenomicRanges[5]. We annotated the consensus peak lists, the common peak set, and DARs to genes using the makeTxDbFromEnsembl function from ChIPseeker (v1.29.1)[46] by associating to each peak/region the nearest gene, setting the TSS region [−1000, 1000] and using them the release 102 from the *mus musculus* Ensemble database. Finally, we selected the chromatin-enriched regions at d4 with annotated nearest genes intersecting some lists of marker genes from the single-cell experiment described[22], (i) the marker genes of endothelial cell clusters in *Tbx1^Cre* Ctrl and cKO embryos at E9.5 (Supplementary Data 5, cluster 6 of the cited publication); (ii) the marker genes of endothelial cell clusters in *Mesp1^Cre* Ctrl and cKO at E9.5 (Supplementary Data 3, cluster 2); (iii) Marker genes of endothelial cell clusters in *Mesp1^Cre* Ctrl and cKO at E9.5 (Supplementary Data 3, cluster 16). We used such regions to identify enriched motifs using HOMER (Hypergeometric Optimization of Motif EnRichment)[47]. From the motif list of regions associated with genes intersecting marker genes of endothelial cell clusters in *Tbx1^Cre*, we selected GATA3 and ERG motifs, we identified those regions containing both motifs and we calculated the percentage. We performed the enhancer prediction of regions associated with genes intersecting the list of marker genes of endothelial cell cluster 6 in *Tbx1^Cre* Ctrl embryos at E9.5, as described in the next section. Finally, we merged the list of regions derived from the intersections of d4 DARs with annotated nearest genes intersecting lists of marker genes of Mesp1cre c2 and c16 clusters and selected unique regions, then we performed the enhancer prediction using these regions.

### Enhancer prediction

We implemented a machine-learning approach to assign a probability score of being enhancers to peak regions from ATAC-seq data using the logistic regression model with the L1 penalty. We performed all analyses using Rstudio and R version 4.2.0 (https://www.r-project.org/).

First, we downloaded the coordinates (chromosome, start-end position) of the 695 enhancers marked as positive from the VISTA ENHANCER Browser (http://enhancer.lbl.gov). Then, since the enhancers' coordinates were in mm9, we mapped them into mm10 using the liftover function (https://genome.ucsc.edu/cgi-bin/hgLiftOver). Next, we created 695 non-enhancer regions to use as negative examples. For this purpose, we randomly shuffled the genome to get genomic coordinates that do not overlap the positive enhancer coordinates with the shuffle function from the BedTools v2.26.0[43] using the parameters -g mm10 -excl the positive enhancer file merged with mm10-blacklist regions file. (https://github.com/Boyle-Lab/Blacklist/blob/master/lists/mm10-blacklist.v2.bed.gz). Such non-enhancer regions have the same lengths as the positive enhancers. Finally, we built a binary response vector with 1390 components assigning 1 to the positive enhancers and 0 to the negative enhancers. Second, we downloaded 385 datasets using Chipseeker (v1.36) R package with the function ChIPseeker::downloadGEObedFiles(genome= "mm10") (https://bioconductor.org/packages/release/bioc/html/ChIPseeker.html) containing peak coordinates of histone modifications, p300 and CTCF transcription factors, at different cell states and cell types (in mm10). After merging the replicates using intersectBed function from

the BedTools, we obtained 327 files. Then, we filtered out the datasets containing KO experiments or other treatments. Overall, we obtained 81 epigenetic tracks of peaks in bed format and 2 in bedgraph format. Next, we intersected the 1390 enhancer and non-enhancer regions with the 81 epigenetic tracks in bed format using the Findoverlaps function from the GenomicRanges R package (v1.52.0) (https://bioconductor.org/packages/release/bioc/html/GenomicRanges.html). Finally, we built a binary matrix of dimension $1390 \times 81$ where we assigned 1 at the positions with an overlap and 0 where they do not. For the remaining two epigenetics tracks in bedgraph format, we used the sum of the coverage at each of the 1390 regions. The feature matrix of dimension $1390 \times 83$ is formed by combining the two parts. We split the dataset into the Training set (80% of the 1390 regions) and the Test-set (20% of the 1390 regions). Then, we trained the L1-penalized Logistic Regression (i.e., Lasso logistic regression) on the training set using K-fold cross-validation to choose the best regularization parameter. To this purpose, we used the glmnet package (v4.1.7) (https://cran.r-project.org/web/packages/glmnet/index.html) with the command: cv.out=cv.glmnet(X.train, Y.train,alpha=1, family = "binomial"). After that, we fitted the training set using the lasso.mod.train = glmnet(X.train, Y.train, lambda = bestlam, alpha = 1, family = "binomial"), where bestlam is the regularization parameter obtained from the cross-validation. In the validation phase, we used the assess.glmnet() function to determine the accuracy of the test-set. Finally, we predicted the scores (i.e., the probability of being an enhancer) to the genomic coordinates of interest. Since the scores vary between 0 (corresponding to minimum probability) and 1 (corresponding to maximum similarity), we used the threshold of 0.5 to establish if a given region is an enhancer. The whole procedure, starting from the random generation of the 695 non-enhancer regions to the final prediction, was repeated 10 times, and the final prediction consisted of the average of the individual prediction scores over which we applied the threshold of 0.5.

### Statistics and reproducibility

All differentiation experiments have been repeated at least three or four times. RNA-seq and ATAC-seq experiments have been performed in two replicates from two independent differentiation experiments. The specific statistical tests applied are specified in the sections above.

For the evaluation of differential gene expression of real-time PCR results, we used the two-way repeated measures ANOVA test. We also used the nonparametric Wilcoxon matched-pairs signed-rank test, one-tailed. For the evaluation of differential gene expression of data from RNA-seq, we used NOIseq (Bayesian test without parametric assumption)[36] and DEseq2 (frequentist test with negative binomial assumption)[37]. For the evaluation of differential accessibility of data from ATAC-seq, we used DiffBind and DEScan2, as detailed above. For enhancer prediction, we used a lasso-type penalized logistic regression test.

### Reporting summary

Further information on research design is available in the Nature Portfolio Reporting Summary linked to this article.

### Data availability

Data supporting the results are included in the figures, Supplementary files, and in the GEO database under the accession number GSE235651 (RNA-seq and ATAC-seq data). The source data behind the graphs in the manuscript can be found in Supplementary Data 6. Any other data and scripts used to run data analysis software are available from the corresponding author (or other sources, as applicable) on reasonable request.

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

## Acknowledgements
We thank Marchesa Bilio for technical help; Laura Pisapia and Enzo Mercadante at the flow cytometry facility, and Salvatore Arbucci at the Microscopy facility of the Institute of Genetics and Biophysics. RNA-seq and ATAC-seq samples were sequenced by Genomix4Life SrL, Salerno, Italy. This work was funded by grants from the Fondazione Telethon GMR22T1012 (to A.B.), Fondation Leducq 15CVD01 (A.B. and E.I.) and the Italian Ministry of University and Research PRIN 20179J2P9J (A.B. and E.I.) and PRIN 2022XFE7M2 (to A.B. and G.L.).

## Author contributions
I.A.: performed experiments, assembled figures, provided conceptual input to experimental design, and edited the manuscript; O.L. and V.P.K.: performed bioinformatic analysis, assembled figures, and contributed to manuscript editing; A.C.: performed experiments; S.A.: performed experiments; G.L.: performed experiments, provided experimental design input, and contributed to manuscript editing; R.F.: performed experiments; C.A.: supervised bioinformatic analysis, provided conceptual input, and contributed to manuscript editing; E.I.: provided funding, contributed conceptual input to experimental design, and edited the manuscript; A.B.: provided funding, designed the experimental plan, and wrote the manuscript.

## Competing interests
The authors declare no competing interests.
