## [Peer Review File · Communications Biology]

Reviewers' comments:

Reviewer #1 (Remarks to the Author):

In their manuscript, Aurigemma et al. seek to identify endothelial regulatory elements that are activated during cardiogenic mesoderm specification using mouse embryonic stem cells. After promoting cardiogenic mesoderm specification, the authors have been able to promote and enhance endothelial differentiation through the addition of VEGFA and Forskolin and they have obtained a relatively pure population of endothelial cells. They have then performed ATAC-seq during endothelial cell specification (and compared them with a list of endothelial marker genes) to localize endothelial specific chromatin regions with a dynamic accessibility~ and assess their likelihood of being enhancers. They have then selected 6 putative endothelial specific enhancers in six genes and have validated their activity thanks to CRISPR-Cas9 epigenetic reprogramming and/or deletion.

This study contains thoughtfully designed experiments and led to the discovery of endothelial-specific enhancers. Overall, this manuscript is important for the field of cardiovascular development.

Please find below specific suggestions for improvement of the manuscript:

1. The title indicates that this study focuses on the cardiopharyngeal lineages. It is clear that the mouse embryonic stem cell differentiation protocol allows cardiogenic mesoderm specification based on the expression of *Mesp1*, *Gata4* and *Gata6* (Figure 1). It would be interesting to further investigate the expression of some cardiopharyngeal mesoderm markers (*Tbx1*, for example, as it has been described for decades by the authors as a key regulator of cardiopharyngeal mesoderm development).
2. Based on the last results and focus on the 6 endothelial specific genes, it would be interesting to add *Notch1* in Figure 1B/C.
3. The authors should maybe comment and discuss the fact that other motifs (T-Box for example) are also found in the *Mesp1*-Cre published dataset, with transcription factors that might be involved in cardiopharyngeal mesoderm development.
4. A more detailed description of the experimental strategy for the CRISPR/Cas9 mediated epigenetic reprogramming or deletion would help the reader appreciate the results. Schematics of the experiments with the different steps of the experiments for example could help or a more detailed description in the text. Even in the methods, the dCas9-LSD1 approach (with ATTO-tagged tracrRNA FACS sorting) is not clear.
5. Can the authors discuss the differences observed between the different Crispr/Cas9 experiments? Notably for *Pecam*, for which the epigenetic reprogramming and deletion gives different results or for the deletion of the enhancer of *Notch1* when *Notch1* targets are differently affected depend on the targeted guid-RNA.

Reviewer #2 (Remarks to the Author):

In this manuscript, Aurigemma et al. integrated several genomic approaches (chromatin accessibility, gene expression, and publicly available single-cell RNA-seq) to answer the identify what are the endothelial regulatory elements activated in early cardiogenic mesoderm. The authors develop a machine-learning strategy to predict the probability of these sequences identified in the integration analysis to be true enhancers.

The first section of the manuscript, the authors focus on analysis of newly generated bulk RNA-seq and ATAC-seq of in vitro cell culture. The authors follow these experiments by integrating lineage-traced embryonic single-cell RNA-seq publicly available. Finally, the authors describe a novel approach using machine learning to identify regulatory elements.

Major concerns:

1. No information provided about samples (N) for bulk RNA-seq experiment present on Figure 1. There is no reference on the N on the figure legend, Methods section or results. As a reader, it is impossible to draw any conclusion without that information provided and, therefore, all subsequent experiments are compromised. Please provide the number of replicates used for this experiment.
2. The authors proposed an unbiased approach to identify new and novel enhancers in EC differentiation. However, they proceed this with an extremely bias selection of enhancer located in known genes involved in that differentiation. None of the six regions selected are within their higher prediction score. To further validate their model, the authors should select higher putative enhancers, preferentially novel ones.
3. The authors demonstrate that loss of the six predicted enhancers indeed change gene expression of a few EC genes. However, there is no functionality assay to confirm if those EC regions are necessary during the differentiation of ECs. Furthermore, it would be of higher interest if authors shown that novel putative enhancers and genes were functionally important in the differentiation of ECs in their model.
4. The title is extremely strong and there is no evidence on the paper to suggest that the authors identify any of the regulatory elements or, in fact, that any of the ECs are necessary to differentiate the cells.

Minor concerns:

Figure 1c: Lack of number of significant genes up or down regulated in the figure and/or legend.

Figure 2b: It would be more impressive if there was either a negative control or show the tube formation assay with cells from d4 and d6 with less capability.

Page 4, ATAC-seq description. Authors used two biological replicates. However, that is the minimum requirement for ENCODE submission and usually only for samples that are hard to obtain (i.e. human

patients or tissues). Is there any reason (biological or otherwise), why there are not more replicates of an in vitro experiment? And are those true biological replicates (two different cell lines) or technical replicates (same cell lines)?

For reference, here is the ENCODE project website: <https://www.encodeproject.org/data-standards/terms/#replication>

Page 4, line 113: "(...) 20268 consensus peak at d2 and 17110 at d8." It should be d4 instead of d8.

Page 4, lines 114-115. No references for "Diffbind" or "Descan2"

Figure 4. Several of the arrows appear to be pointing at wrong spots. No scale on Y axis makes it hard to understand what level of fold change or opening we are observing.

Figure 5 and 6. Relative gene expression of several genes seem extremely low (ranges from 0.01-0.06). Is there an error on this calculation or are these values this low? IF this low, are they biologically meaningful?

Figure 4 and 6a. Poor resolution even on PDF and open on computer.

Reviewers' comments:

Reviewer #1 (Remarks to the Author):

1. The title indicates that this study focuses on the cardiopharyngeal lineages. It is clear that the mouse embryonic stem cell differentiation protocol allows cardiogenic mesoderm specification based on the expression of *Mesp1*, *Gata4* and *Gata6* (Figure 1). It would be interesting to further investigate the expression of some cardiopharyngeal mesoderm markers (*Tbx1*, for example, as it has been described for decades by the authors as a key regulator of cardiopharyngeal mesoderm development).

Answer:

We have added *Tbx1* gene expression RT-PCR to the panel of genes shown in Fig. 1B. *Tbx1* is first detectable at d4, still detectable during endothelial differentiation (d6), but no longer detectable in the near-homogeneous endothelial population (d8).

2. Based on the last results and focus on the 6 endothelial specific genes, it would be interesting to add *Notch1* in Figure 1B/C.

Answer:

We have added *Notch1* to Figure 1B and C. Note that we had mistakenly indicated that *Notch1* was differentially expressed between d2 and d4. It is upregulated during EC differentiation (d6-d8), although the enhancer tested is opened between d2 and d4. This is now corrected on Tab. 3.

3. The authors should maybe comment and discuss the fact that other motifs (T-Box for example) are also found in the *Mesp1-Cre* published dataset, with transcription factors that might be involved in cardiopharyngeal mesoderm development.

Answer:

That is an interesting observation that should be addressed experimentally in the future. We have added to the Discussion: " Interestingly, the *Mesp1^{cre}* dataset motifs also included transcription factor families that play a role in CPM development, such as T-BOX, FOX, and MEIS factors (Additional file 4: Tab. S4), raising the question of whether they may be involved in enabling the EC transcription program in the CPM. "

4. A more detailed description of the experimental strategy for the CRISPR/Cas9 mediated epigenetic reprogramming or deletion would help the reader appreciate the results. Schematics of the experiments with the different steps of the experiments for example could help or a more detailed description in the text. Even in the methods, the dCas9-LSD1 approach (with ATTO-tagged tracrRNA FACs sorting) is not clear.

Answer:

We have extended the description in the methods section and we also included a cartoon of the experimental procedure on Fig. 5A, lower panel.

5. Can the authors discuss the differences observed between the different Crispr/Cas9 experiments? Notably for *Pecam*, for which the epigenetic reprogramming and deletion gives different results or for the deletion of the enhancer of *Notch1* when *Notch1* targets are differently affected depend on the targeted guide-RNA.

Answer:

We have added to the Discussion: " Epigenetic reprogramming, while providing consistent results, proved to be variable in our hands. Sources of variability may be the efficiency of transfection, the gRNAs, or perhaps variable extent of chromatin modification induced by the dCAS9:LSD1 complex. Furthermore, the inconsistent results obtained with the *Pecam1* putative enhancer using epigenetic reprogramming and gene editing may be due to different reasons. We speculate that perhaps the sequence is not a regulatory element (as suggested by the low prediction score) but the genetic deletion may have altered the expression of the gene by interfering with processes like RNA maturation/splicing or causing other structural perturbation of the gene."

Reviewer #2 (Remarks to the Author):

Major concerns:

1. No information provided about samples (N) for bulk RNA-seq experiment present on Figure 1. There is no reference on the N on the figure legend, Methods section or results. As a reader, it is impossible to draw any conclusion without that information provided and, therefore, all subsequent experiments are compromised. Please provide the number of replicates used for this experiment.

Answer:

For ATAC-seq and RNA-seq, we have used 2 replicates/each. We apologize for omitting the information that is now included in the revision (figure legend, results, and methods sections). The replicates come from two independent differentiation experiments, therefore they are biological replicates. We only have one mES cell line.

We are aware that 2 replicates is the minimum. However, we have taken precautions and we have considered only DARs or DEGs that are significant in both replicates. In addition, DEGs were calculated with two different algorithms and we only considered genes differentially expressed in both. This approach is restrictive and in part compensates for the low number of replicates. Unfortunately, cost is also an issue for us.

2. The authors proposed an unbiased approach to identify new and novel enhancers in EC differentiation. However, they proceed this with an extremely bias selection of enhancer located in known genes involved in that differentiation. None of the six regions selected are within their higher prediction score. To further validate their model, the authors should select higher putative enhancers, preferentially novel ones.

Answer:

The "candidate gene" approach (we respectfully disagree with the definition "extremely biased") is actually unbiased to the predictive score, this is why we have essentially ignored it for this part of the study, although we have calculated it.

We regret that we cannot do additional dCAS9 experiments because of serious time constraints that we have in the lab, exacerbated by the extremely long time that has been required to get the first review of the manuscript (70 days!). We have discussed this issue with the Editor and decided it is not necessary to address this concern with additional data. The fact that several of the predicted enhancers have actually been validated in the literature (15 out of the 57 sequences with score above 0.5, i.e. 26%) is per se a strong evidence that the method is effective; there are not many EC enhancers published.

To the Discussion we have added: "Of the 57 putative enhancers scoring > 0.5 identified through our unbiased approach, 15 (26%) were already reported in the literature, thus

suggesting that our approach was efficient in identifying likely regulatory elements in our model. In addition, motif analysis showed enrichment of transcription factors known to be involved in EC development, further supporting the suitability of the approach. However, we did not validate these putative enhancer directly in our model, thus further work will be necessary to establish the reliability of our approach for systematic identification of cell type-specific enhancer sequences."

Also, in the Conclusion we state: "more extensive bench-validation experiments will be required before the approach may be considered an established pipeline for enhancer identification."

3. The authors demonstrate that loss of the six predicted enhancers indeed change gene expression of a few EC genes. However, there is no functionality assay to confirm if those EC regions are necessary during the differentiation of ECs. Furthermore, it would be of higher interest if authors shown that novel putative enhancers and genes were functionally important in the differentiation of ECs in their model.

Answer:

Gene expression is a functional assay for enhancers. It was not our intention to demonstrate that these enhancers are *required* for differentiation, which is unlikely for an individual enhancer. We did demonstrate that they affect regulation of their respective genes in 5 of six cases tested, especially in the later stages of the differentiation protocol (d6-d8), thus suggesting that they may be important in post differentiation processes.

To further address the point of the reviewer, and in compliance with a request from the Editor, we have included two additional sets of experiments using the *Notch1* enhancer deletion lines. In the first set of experiments we produced gastruloids from the same ES cell lines and observed the endothelial networks developed within them. In the second set of experiments, we performed Matrigel assays of mutant and WT differentiated cells. Results are shown in the new Fig. 7. Gastruloid images show apparently higher density of the EC (Pecam1+) network in mutant compared to wt cells. We could not quantify these data due to 3D complexity of gastruloids. Therefore, we quantified 2-dimensional matrigel data that, consistently, indicated significantly increased branching of the EC network. These results are consistent with previously reported function of *Notch1* in suppressing branching.

4. The title is extremely strong and there is no evidence on the paper to suggest that the authors identify any of the regulatory elements or, in fact, that any of the ECs are necessary to differentiate the cells.

Answer:

The new title is: "Endothelial gene regulatory elements *associated with* cardiopharyngeal lineage differentiation".

Contrary to the reviewer's statement, we have validated 5 regulatory elements that are activated in the process under study. We also identified a substantial list of highly likely regulatory elements, 26% of which were validated in the literature.

As clarified above, our aim is not to identify enhancers "necessary to differentiate the cells". For example, the *Notch1* enhancer that we have targeted is involved in the morphogenesis of vessels rather than EC differentiation.

Minor concerns:

- Figure 1c: Lack of number of significant genes up or down regulated in the figure and/or legend.

Answer

Numbers are now indicated in the legend.

- Figure 2b: It would be more impressive if there was either a negative control or show the tube formation assay with cells from d4 and d6 with less capability.

Answer

We have added a negative control (d4 cells) to Fig. 2B.

- Page 4, ATAC-seq description. Authors used two biological replicates. However, that is the minimum requirement for ENCODE submission and usually only for samples that are hard to obtain (i.e. human patients or tissues). Is there any reason (biological or otherwise), why there are not more replicates of an in vitro experiment? And are those true biological replicates (two different cell lines) or technical replicates (same cell lines)?

For reference, here is the ENCODE project website: <https://www.encodeproject.org/data-standards/terms/#replication>

Answer:

We only have one parental ES cell line. The differentiation process is the source of variability and having two independent differentiation experiments should qualify as two biological replicates (as indicated also in the web page cited by the reviewer). In addition, we only used consensus results from the two experiments, rather than their union.

- Page 4, line 113: "(...) 20268 consensus peak at d2 and 17110 at d8." It should be d4 instead of d8.

Answer:

Corrected, thanks.

Page 4, lines 114-115. No references for "Diffbind" or "Descan2"

Answer:

The citations have been added also to the results section (they were in the Methods).

- Figure 4. Several of the arrows appear to be pointing at wrong spots. No scale on Y axis makes it hard to understand what level of fold change or opening we are observing.

Answer:

We removed the arrows and replaced them with boxes. We have also added the range of values.

- Figure 5 and 6. Relative gene expression of several genes seem extremely low (ranges from 0.01-0.06). Is there an error on this calculation or are these values this low? IF this low, are they biologically meaningful?

Answer:

The expression level is calculated using the $2^{(-\Delta Ct)}$ method from 4 or 5 different experiments. We have no reason to believe that they are meaningless even though they are low.

- Figure 4 and 6a. Poor resolution even on PDF and open on computer.

Answer:

We replaced all of the screenshots.

Response to Editor's comments:

1) Since dCas9-recruited repressors could potentially lead to broader heterochromatin spreading, if a putative enhancer is a few kb from the promoter, it's possible that any observed repression is just an effect of dCas9 acting on the promoter itself. Therefore, we ask that you acknowledge this point in your revised manuscript and discuss the distance between each putative enhancer element and the relevant gene promoter (rather than only including this information in Table 2).

Answer:

We have added the following statement (Discussion):

"The dCas9-recruited repressor could potentially cause chromatin modifications beyond the intended targeted sequence, and cause gene downregulation, particularly if the promoter is nearby. The six enhancers tested with this method are all fairly distant from the transcription start site (TSS, Tab. 3). The closest is 8.7Kb from the TSS, the others are between 15 and 67 Kb. "

2) revise the text to de-emphasize this as a methods/pipeline paper and clearly explain your rationale for using a targeted approach for validation of enhancers.

Answer:

We have added in the Discussion: "Of the 57 putative enhancers scoring > 0.5 identified through our unbiased approach, 26% were already known in the literature, thus suggesting that our approach was efficient in identifying likely regulatory elements in our model. In addition, motif analysis showed enrichment of transcription factors known to be involved in EC differentiation. However, we did not validate these putative enhancer directly in our model, thus further work will be necessary to establish the reliability of our approach for systematic identification of enhancer sequences."

" The candidate gene approach was designed to identify regulatory sequences "activated" in our model and associated with genes known to be involved in EC differentiation. We have validated them regardless of the prediction score and found five of the six tested putative REs to regulate the respective genes. "

In the Conclusion we have added: " The identification and validation strategies applied here are applicable to other cell types, whenever a suitable differentiation model is available, although more extensive bench validation experiments will be required before the approach may be considered an established pipeline for enhancer identification."

REVIEWERS' COMMENTS:

Reviewer #1 (Remarks to the Author):

Aurigemma et al. have submitted a revised version of their manuscript untitled "Endothelial gene regulatory elements associated with cardiopharyngeal lineage differentiation". They have added few figure panels which have improved their manuscript.

The authors have addressed or discussed most of our comments.

Reviewer #2 (Remarks to the Author):

I have thoroughly reviewed the revisions made by the authors in response to the comments I had previously provided. It is clear that they have taken the time to carefully consider each point that was raised. They have made changes to the manuscript that adequately address all of my concerns. Their revisions have not only resolved the issues I pointed out, but also enhanced the overall quality of the paper. Therefore, I am satisfied with the modifications they have made in response to my comments.